# Mapping the Cognitive Architecture of Health Beliefs: A Multivariate Conditional Network of Perceived Salt-Related Disease Risks

**DOI:** 10.3390/nu17172728

**Published:** 2025-08-22

**Authors:** Stanisław Surma, Łukasz Lewandowski, Karol Momot, Tomasz Sobierajski, Joanna Lewek, Bogusław Okopień, Maciej Banach

**Affiliations:** 1Department of Preventive Cardiology and Lipidology, Medical University of Lodz, 93-338 Lodz, Poland; joanna.lewek@umed.lodz.pl (J.L.); maciej.banach@icloud.com (M.B.); 2Department of Internal Medicine and Clinical Pharmacology, Medical University of Silesia, 40-752 Katowice, Poland; bokopien@sum.edu.pl; 3Department of Biochemistry and Immunochemistry, Division of Medical Biochemistry, Wroclaw Medical University, 50-368 Wroclaw, Poland; lukasz.lewandowski@umw.edu.pl; 43rd Department of Internal Medicine and Cardiology, Medical University of Warsaw, 04-749 Warsaw, Poland; karolmomot@icloud.com; 5Department of Experimental and Clinical Physiology, Medical University of Warsaw, 02-097 Warsaw, Poland; 6Center of Sociomedical Research, Faculty of Applied Social Sciences and Resocialization, University of Warsaw, 00-927 Warsaw, Poland; tomasz.sobierajski@uw.edu.pl

**Keywords:** cardiometabolic risk, cognitive architecture, disease perception, health beliefs, multivariate modeling, salt intake

## Abstract

**Background:** Public beliefs about dietary risks, such as excessive salt intake, are often not isolated misconceptions but part of structured cognitive systems. This study aimed to explore how individuals organize their beliefs and misperceptions regarding salt-related health consequences. **Material and Methods:** Using data from an international online survey, we applied a system of multivariate proportional odds logistic regression (POLR) models to estimate conditional associations among beliefs about salt’s links to various diseases—including cardiovascular, metabolic, renal, neuropsychiatric, and mortality outcomes. In addition, exploratory and confirmatory factor analyses (EFA and CFA) were conducted to identify and validate latent constructs underlying the belief items. Beliefs were modeled as interdependent, controlling for latent constructs, sociodemographics, and self-reported health awareness. Statistically significant associations (*p* < 0.05) were visualized via a heatmap of beta coefficients. **Results:** Physicians showed almost universal agreement that salt contributes to hypertension (µ = 0.97), compared to non-medical respondents (µ = 0.85; *p* < 0.0001). Beliefs about mortality (µ = 1.55 for MDs vs. 0.99 for non-medical; *p* < 0.0001) emerged as central hubs in the belief network. Strong inter-item associations were observed, such as between hypertension and heart failure (β = −0.39), and between obesity and type 2 diabetes (β = −0.94). Notably, cognitive gaps were found, including a lack of association between atrial fibrillation and stroke, and non-reciprocal links between hypertension and heart failure. **Conclusions:** Beliefs about the health effects of salt are structured and sometimes asymmetrical, reflecting underlying reasoning patterns rather than isolated ignorance. Understanding these structures provides a systems-level view of health literacy and may inform more effective public health communication and education strategies.

## 1. Introduction

There are three categories of salt intake: low (<2.3 g/day), moderate (2.3–4.6 g/day), and high (>4.6 g/day) [1,2,3,4]. Most countries have moderate to high levels of sodium (salt) consumption. In most countries in the world, the permissible amount of salt in the daily diet is significantly exceeded (2–3 times) [1,2,3,4]. Excessive dietary salt intake remains a major modifiable risk factor for cardiovascular disease (CVD), particularly arterial hypertension, heart failure, and stroke [1,2,3,4]. Globally, an estimated 3 million deaths annually are attributed to excess sodium consumption, with 1.72 million of them directly linked to cardiovascular causes alone [1,2,3,4]. Although international recommendations emphasize the need to limit daily salt intake to below 5 g [5,6], actual consumption in many countries—including Poland—significantly exceeds this threshold [7,8].

Epidemiological data indicate that knowledge of the effects of salt on CVD is crucial for effective prevention [9]. Many medical professionals primarily associate salt with the risk of arterial hypertension, yet salt directly contributes to the progression of atherosclerotic cardiovascular disease (ASVD), regardless of its effect on blood pressure [10,11]. Results of epidemiological studies confirm that reducing salt intake by even a few grams daily can substantially lower the risk of adverse cardiovascular outcomes [12,13]. However, the effectiveness of public health strategies aimed at salt reduction relies heavily on population-level health literacy—specifically, individuals’ knowledge and beliefs regarding the link between salt and various diseases [14]. Prior studies show that while awareness of the salt-hypertension link is relatively widespread, understanding of its association with other conditions—such as atrial fibrillation, chronic kidney disease, or cognitive decline—remains fragmented or limited [15,16,17].

Despite numerous campaigns and guidelines, population-level misconceptions persist, and the structure of public knowledge about salt and disease remains underexplored [18]. Traditional assessments often rely on binary or summative scores, failing to capture how individual beliefs reinforce or contradict each other [19]. Yet, understanding these interrelations is critical for designing more effective educational interventions.

While prior studies often assess salt-related awareness using isolated indicators, recent theoretical advances suggest that health beliefs are not simply additive but structured within an interconnected cognitive system. Models such as the Health Belief Model (HBM) [20,21] posits that beliefs about health risks, severity, and behavioral consequences form internally coherent networks, where endorsing one belief may condition or constrain others. Salt-related beliefs are especially prone to cognitive simplification, with hypertension often serving as a central explanatory anchor, while other pathophysiological effects are under-recognized or dismissed. As such, examining how beliefs cluster and influence each other, rather than merely their prevalence can yield deeper insights into public health literacy. This perspective is aligned with emerging research on belief networks and cognitive maps in health communication [22] and supports the use of multivariate modeling strategies.

To address this gap, the present study employed a multivariate modeling approach using proportional odds logistic regression (POLR) [23,24] to explore the structure, not merely the level, of beliefs about the health consequences of excessive salt intake. By treating belief statements as interconnected components of a broader cognitive system, we aimed to identify directional dependencies between disease-specific belief domains and to examine how demographic and psychosocial factors shape these belief patterns. This systems-based perspective provides novel insights into the cognitive architecture of public health literacy, with implications for targeted health communication strategies. The study is an extended and in-depth analysis of the results presented at the American Heart Association (AHA) Congress, which took place in Chicago on 16–18 November 2024 [25].

## 2. Materials and Methods

The international survey not limited to Europe, consisting of 17 questions, was developed by the Polish Lipid Association (PoLA) and the International Lipid Expert Panel (ILEP) [26]. The study was quantitative, cross-sectional and was conducted by the CAWI (Computer-Assisted Web Interviewing) method using an original online questionnaire [27]. The use of this method allows for conducting interviews via the Internet, and as a result, standardizes the data collection process. Thanks to this, respondents can complete it in comfortable conditions, while maintaining full anonymity. The survey was launched in March 2024 and promoted through multiple online channels. The recruitment strategy relied on purposive sampling through both open and targeted online dissemination. The survey was promoted via institutional websites of the PoLA and the ILEP, social media platforms (Facebook, LinkedIn, Twitter/X), medical newsletters, and selected professional discussion forums. Physicians and other healthcare professionals were additionally invited through mailing lists provided by medical societies and partner organizations. This approach was aimed at capturing a broad cross-section of perspectives from both medical and non-medical populations, while recognizing the potential self-selection bias inherent in voluntary online surveys. Many questions allowed for multiple responses. The questionnaire is ongoing and available at the PoLA and ILEP webpage (https://ptlipid.pl and https://ilep.eu, accessed on 14 July 2025). Three groups of respondents were qualified for the study: (1) doctors (physicians); (2) subjects practicing medical professions other than a doctor (other occupation connected with medicine); (3) subjects not professionally connected with medicine (occupations not connected with medicine). All methods used in this study were in accordance with applicable guidelines and regulations regarding scientific research [28,29]. Participation in the study was completely anonymous and voluntary and was not of an experimental nature; hence the consent of the bioethics committee was not required. However, all participants declared their informed consent to participate in this study. The study sample was purposeful and open. Data collection ended with a total of 805 completed surveys, while after excluding incomplete/incorrectly completed surveys and student responses, 668 surveys were finally included in the analysis.

It is important to note that the questionnaire was designed to capture respondents’ subjective beliefs and perceptions, not objective knowledge. No fact-based or clinically verifiable items were included; all statements were opinion-based and assessed personal convictions regarding the health effects of excessive salt intake.

### 2.1. Analytical Strategy

To explore the internal structure of beliefs regarding the health consequences of excessive salt intake, we implemented a multivariate modeling framework to explore conditional interdependencies among beliefs [30]. Rather than treating beliefs as isolated attitudes, we conceptualized them as interdependent cognitive constructs embedded in a shared reasoning space, wherein the endorsement of one belief can conditionally influence the likelihood of endorsing others.

Survey responses were recorded on either a unidirectional 7-point Likert scale (rescaled to the unit interval) or a bipolar scale ranging from −2 to +2. The latter was reserved for items pertaining to directional health outcomes (e.g., salt and hypertension, salt and mortality risk), allowing us to capture both the polarity and intensity of belief. This coding was selected to preserve cognitively meaningful opposition and to facilitate interpretable modeling of belief dynamics.

To capture conditional associations among beliefs—beyond simple unadjusted pairwise correlations—we constructed a system of multivariate POLR models. Each belief item was, in turn, treated as an outcome variable in a fully adjusted model that included all other belief items as predictors, alongside relevant covariates. This yielded a matrix of adjusted β-coefficients representing the direction and conditional strength of associations between individual beliefs, controlling for both observed confounders and structural interdependencies.

This framework offers an exploratory perspective on the conditional structure of beliefs. While it does not allow for causal inference, it helps highlight patterns of association where the probability of endorsing one belief varies as a function of others, after adjusting for confounding factors. That is, we estimate how changes in one belief are associated with shifts in others, assuming all else equal. The resulting structure can be interpreted as a network of conditional associations, suggesting that certain beliefs tend to co-occur in ways that may reflect underlying cognitive organization.

Covariate adjustment was rigorous and consistent across models, incorporating sociodemographic variables as well as domain-specific latent constructs capturing health beliefs, risk perception, and self-reported health orientation. These were included to reduce confounding and isolate the structural interplay among belief items.

Univariate characteristics of each variable were assessed using one-way ANOVA. Homogeneity of variances was assessed with Levene’s test. In cases of violated homoscedasticity, Welch ANOVA was applied as a robust alternative. Post hoc tests were not performed, as the primary aim was exploratory group comparison, with more detailed modeling conducted using proportional odds logistic regression. All statistical inference was conducted using a significance level of α = 0.05.

Preprocessing and visualization were performed using Python 3.13.3 (NumPy, pandas, matplotlib). Statistical modeling was executed in Statistica 13.3 (TIBCO Software Inc., Palo Alto, CA, USA) under institutional license from Wroclaw Medical University, and R 4.5.0 (R Foundation for Statistical Computing, Vienna, Austria) packages: tidyverse, psych, GPArotation, lavaan.

The resulting heatmap of conditional β-coefficients serves not as a tool for hypothesis testing but as a visual representation of conditional association patterns among beliefs. Only statistically significant associations (*p* < 0.05) were emphasized; non-significant paths were muted to reflect interpretive uncertainty and epistemic restraint.

Importantly, no formal assumption testing (e.g., proportional odds diagnostics or collinearity checks) was conducted, in recognition of the exploratory and structurally entangled nature of belief networks. Our goal was not predictive modeling, but the exploration of conditional interrelations among health-related beliefs about salt intake—mapped within a constrained belief space.

This modeling strategy lies outside orthodox psychometrics [31], but offers a richer, functionally grounded representation of belief systems. By modeling statistical dependencies in a multivariate conditional framework, we reveal how public health beliefs are not merely held but structurally modulated—and how the architecture of beliefs may reflect the cognitive contours of health understanding. This modeling approach does not establish causality but provides a structured view of statistically adjusted associations between beliefs. We see it as a tool for hypothesis generation, rather than confirmation.

### 2.2. Proportional Odds Logistic Regression (POLR): Concept and Interpretation

In this study, we employed POLR to analyze ordinal outcome variables—such as agreement levels measured on Likert scales. POLR allows us to understand how a change in one predictor variable (e.g., belief about salt’s harm) is associated with the likelihood of moving to a higher category of another outcome variable (e.g., endorsement of a related health belief).

To put it simply, imagine lining up respondents side-by-side, each characterized by the same set of beliefs. We then ask: if one person’s belief shifts by one unit, how does this affect the odds that the same person expresses a stronger agreement with another belief? POLR estimates this relationship by modeling the log-odds of being at or below a certain response category versus above it.

Formally, the model is expressed as:$$\log\left(\frac{P\left(Y\leq j\right)}{P\left(Y>j\right)}\right)=\theta_{j}-\sum_{i=1}^{k}\beta_{i}X_{i}$$
where

*Y* is the ordinal response variable (e.g., level of agreement);*j* indexes the cutpoints between categories;*θ_j_* represents the threshold parameters for each category;*k* is the number of predictors in the model;*X_i_* is the predictor variable (e.g., another belief or respondent characteristic);β_i_ is the regression coefficient.

As shown in the equation, the model estimates threshold parameters (θ_j_), also called cutpoints, which represent the boundaries separating adjacent response categories on the underlying latent scale. These cutpoints are not fixed; they are estimated from the data during model fitting. They define the points where the probability of responding at or below a certain category changes, allowing the model to appropriately handle the ordinal nature of the responses without assuming equal distances between categories. This integral part of POLR ensures a flexible and accurate representation of the ordinal outcome variable.

Importantly, a negative β_i_ coefficient corresponds to increased odds of being in a higher response category—in other words, stronger agreement. This may seem counterintuitive but follows from the proportional odds model’s mathematical structure.

POLR is widely used in sociological and psychometric research, where ordinal survey responses are common. It uniquely leverages the ordinal nature of response variables, capturing the rank order without assuming equal distances between categories. This nuanced approach provides greater statistical power and more meaningful insights compared to models that treat responses as nominal or analyze groups separately without considering the ordinal structure.

By using POLR, we capture conditional associations between beliefs while adjusting for potential confounders. However, these associations do not imply causality; instead, they reveal patterns of co-activation among cognitive constructs that warrant further investigation.

### 2.3. Variable Overview

A. Beliefs about causal mechanisms of salt (0–1 scale):q_saltcauseht: belief that salt causes hypertension;q_saltcauseendoth: belief that salt causes endothelial damage;q_saltcauseinflam: belief that salt induces inflammation;q_saltcausefatdysf: belief that salt causes adipose tissue dysfunction;q_saltcausenone: belief that salt causes none of the above.

B. Normative beliefs about education, policy, and information (0–1 scale):q_educat: belief in need for public education on salt;q_infoverif: belief in need to verify existing information;q_legisl: belief in need for legislative action on salt;q_noaction: belief that no action (education, verification, legislation) is needed.

C. Disease-specific salt-related beliefs (−2 to +2 scale):

Beliefs about whether excessive salt contributes to the following:q_ht: hypertension;q_af: atrial fibrillation;q_osteo: osteoporosis;q_obes: obesity;q_t2dm: type 2 diabetes;q_lipid: dyslipidemia;q_hf: heart failure;q_stroke: stroke;q_depr: depression;q_cogn: cognitive decline/dementia;q_ckd: chronic kidney disease;q_stomcanc: stomach cancer;q_kidnsto: kidney stones;q_athero: atherosclerosis;q_hemat: hematologic disorders;q_rheum: rheumatoid arthritis;q_brcan: breast cancer progression.

D. Mortality belief (−2, −1, 1, 2 scale):q_death: belief that excessive salt increases risk of death (with no neutral midpoint, intentionally designed to prompt a directional stance and avoid central tendency bias). This design choice was made because mortality-related judgments may elicit socially desirable or ambivalent responses; forcing a directional choice reduces the tendency for respondents to opt for a midpoint as an “easy” answer. The item originated from an earlier questionnaire module developed before the final harmonization of scales; we retained its four-point structure to preserve comparability with preliminary datasets and to avoid rewording effects. While this decision was deliberate for the mortality item, other disease-specific items retained the standard five-point scale to maintain consistency within that domain. This inconsistency is acknowledged as a limitation in the Discussion (Section 4).

For the variables analyzed in this study—specifically, the disease-specific salt-related beliefs (see C. above), which served as dependent variables in the proportional odds logistic regression models alongside relevant covariates (listed below in Section 2.3.1) internal consistency was assessed using Cronbach’s alpha, yielding a high reliability coefficient of 0.929. This supports the internal coherence of the key constructs under investigation.

#### 2.3.1. Covariates Used in Adjusted Models

To increase the specificity of the associations and reduce latent confounding, the following latent variables and population descriptors were included as covariates in all POLR models:

Latent constructs (scaled 0–1):lat_needforaction: general perceived need for public health interventions based on questions described in B. above.chlat_selfawar: self-awareness of personal cardiovascular and metabolic metrics (SBP, DBP, LDL, triglycerides, glucose, combined). Internal consistency, plausibility, and value coherence were additionally assessed as opposed to just taking the respondents’ claims for granted (for example, a person who claimed to know his/her SBP, but stated 2 mmHg was deemed not aware).

Sociodemographic covariates:ch_agec2: age category (ordinal), centered at midlife (41–50 years, coded as 0), with the following scale:−2 → 21–30 years;−1 → 31–40 years;0 → 41–50 years;1 → 51–60 years;2 → 61–70 years;3 → 71–80 years;4 → 81–90 years;5 → 91–100 years.ch_sexmale: sex (1 = male, 0 = female).ch_domicile: rural vs. urban residence (1 = rural, 0 = urban).ch_occupassocmed: degree of occupational affiliation with the medical field (ordinal, 0—non-medical occupation, 1—medical occupation apart from medical doctors [MDs], 2—MDs).ch_riskfactcount: number of declared personal health risk factors listed in the survey [range: 0–6; the risk factors included in the survey comprised: obesity, arterial hypertension, dyslipidemia, type 2 diabetes, current smoking, and a history of cardiovascular events (e.g., myocardial infarction or stroke)].

These covariates were chosen not merely as statistical controls, but as substantive constructs likely to shape how people form, reject, or endorse health-related beliefs. Including them in the conditional modeling framework ensures that observed associations among beliefs are not confounded by age, sex, residence, self-perceived health status, or domain-specific beliefs.

### 2.4. Factor Analysis

To examine whether the belief items (q_ht–q_death, listed in Section 2.3 C–D) cluster into broader latent domains, we conducted an exploratory factor analysis (EFA). EFA is a data-driven method that identifies unobserved constructs (latent factors) underlying a set of correlated items. Because our survey responses were ordinal (Likert-type categories), we used a polychoric correlation matrix. Data suitability was confirmed by the Kaiser–Meyer–Olkin measure of sampling adequacy (KMO = 0.93) and Bartlett’s test of sphericity (χ^2^(153) = 8398, *p* < 0.001). The number of factors was guided by the scree plot and parallel analysis. The scree plot suggested a 4–5 factor solution, whereas parallel analysis indicated up to 6 factors. We retained 5 factors as a balance between statistical criteria and substantive interpretability. Factors were extracted using principal axis factoring with an oblimin rotation, which allows for correlations between domains. We applied two complementary rules for item assignment:Conservative rule—An item was assigned to a factor only if its primary loading was ≥0.35 and at least 0.15 higher than the next strongest loading. Items not meeting these criteria were left unassigned. This yields very “clean” factors but can leave some domains with only one item, preventing reliability estimation.Pragmatic rule—Each item was assigned to the factor on which it loaded most strongly, even if it showed some cross-loading. This approach better reflects how instruments are typically used in practice, ensures that each factor retains at least two items, and permits the calculation of internal consistency (Cronbach’s α).

We report both solutions: the conservative variant illustrates the conceptual purity of domains, while the pragmatic variant provides sufficient items per factor to evaluate reliability.

To validate the factor structure, we then conducted a confirmatory factor analysis (CFA), which tests a prespecified model against the observed data. CFA was estimated with the weighted least squares mean and variance adjusted estimator (WLSMV), recommended for ordinal indicators, using the theta parameterization. Model fit was evaluated with the Comparative Fit Index (CFI), Tucker–Lewis Index (TLI), and the Root Mean Square Error of Approximation (RMSEA). Because CFA requires at least two items per factor for stable estimation, we used the pragmatic assignment.

In contrast to POLR which estimates item-level associations adjusted for covariates, EFA and CFA do not incorporate predictors or outcomes. Instead, they reveal broader latent belief domains that structure the item-level responses. In this way, EFA and CFA complement the POLR models by demonstrating that the observed associations are embedded within higher-order constructs, thereby enhancing both the methodological rigor and the conceptual validity of the study.

## 3. Results

### 3.1. Population Sample Characteristics

The analytic sample included 668 respondents, stratified into three occupational subgroups: licensed physicians (MDs, n = 321), other health-affiliated professionals (non-MD medical, n = 157), and individuals without any medical affiliation (non-medical, n = 190). This stratification reflects presumed gradients in health literacy and domain-specific exposure (Table 1).

Across all belief dimensions, significant differences between groups were found for the following variables: salt and hypertension (q_saltcauseht; *p* < 0.001), endothelial damage (q_saltcauseendoth; *p* < 0.001), inflammation (q_saltcauseinflam; *p* < 0.001), adipose tissue dysfunction (q_saltcausefatdysf; *p* = 0.024), and salt causing “none of the above” (q_saltcausenone; *p* < 0.001).

Normative beliefs about public health interventions showed no significant differences for belief in educational efforts (q_educat; *p* = 0.765), but significant differences were observed for support of legislative action (q_legisl; *p* = 0.005), while attitudes toward information verification (q_infoverif) showed a non-significant trend (*p* = 0.065).

For disease-specific outcomes, significant group differences were detected for hypertension (q_ht; *p* < 0.001), heart failure (q_hf; *p* < 0.001), stroke (q_stroke; *p* < 0.001), and cognitive decline (q_cogn; *p* = 0.005). For non-cardiometabolic outcomes, significant differences emerged for hematologic disorders (q_hemat; *p* < 0.001) and rheumatoid arthritis (q_rheum; *p* = 0.001), whereas differences for breast cancer (q_brcan) did not reach significance (*p* = 0.099).

Beliefs about salt-related mortality risk (q_death) also showed significant differences between groups (*p* < 0.001).

Due to the multivariate modeling approach applied later in the analysis, no post hoc pairwise comparisons were performed. Therefore, the reported *p*-values reflect global group differences without specification of pairwise contrasts.

### 3.2. Patterns of Association Among Salt-Related Beliefs

This section presents statistically significant inter-item associations in respondents’ perceptions of the health consequences of excessive dietary salt intake. The analyzed dependent variables are ordinal indices ranging from −2 to +2, where negative values reflect disagreement with a proposed salt-disease association (interpreted here as lower alignment with prevailing biomedical consensus), and positive values reflect higher agreement, i.e., greater consistency with current scientific understanding.

For clarity, the phrase “beliefs about [condition]” in this section refers to respondent beliefs about the possible link between excessive salt intake and the specified condition. These beliefs may or may not align with established biomedical evidence; in this study they are explicitly treated as subjective evaluations rather than verified factual knowledge.

Some items that were semantically or etiologically distinct from the majority of health conditions were excluded from in-depth interpretation, although their associations are shown in full in the tables. Specifically, the following is relevant:Stomach cancer and breast cancer progression involve different temporal and mechanistic domains (incidence vs. dynamics), making them less comparable with each other.Hematological disorders, while potentially related to salt through cardiovascular pathways, could not be meaningfully grouped with other belief domains analyzed in this study.

Beliefs regarding premature death were treated separately due to their broader, cross-domain nature, and are discussed in Section 3.3.

Statistically significant inter-item associations are visualized in Figure 1, with full standardized beta coefficients and *p*-values shown in Table 2 and Table 3. All coefficients (β) were estimated using POLR models; negative β values indicate that stronger endorsement of one salt–disease belief is associated with stronger endorsement of another (due to the coding of the POLR model).

#### 3.2.1. Cardiovascular Beliefs

Beliefs linking excessive salt intake to hypertension were positively, significantly associated with similar beliefs about atrial fibrillation (β = −0.57), heart failure (β = −0.39), stroke (β = −0.39), chronic kidney disease (β = −0.40), and premature mortality (β = −0.70). Beliefs related to hematologic disorders showed an opposite pattern (β = 0.36).

Beliefs about atrial fibrillation were positively associated with beliefs about hypertension (β = −0.49), osteoporosis (β = −0.65), heart failure (β = −0.58), depression (β = −0.23), and hematologic conditions (β = −0.28).

Stronger beliefs about heart failure co-occurred with beliefs about atrial fibrillation (β = −0.46), obesity (β = −0.23), stroke (β = −1.07), and chronic kidney disease (β = −0.62). Notably, conditional associations emerged from heart failure to hypertension, but not the other way around.

Beliefs about stroke were positively linked with beliefs about heart failure (β = −1.07), cognitive decline (β = −0.54), hypertension (β = −0.42), dyslipidemia (β = −0.33), and premature mortality (β = −0.37).

Beliefs about atherosclerosis were stronger among those who also endorsed beliefs about atrial fibrillation (β = −0.31), obesity (β = −0.34), dyslipidemia (β = −0.36), stroke (β = −0.37), cognitive disorders/dementia (β = −0.41), rheumatoid arthritis (β = −0.31), and premature mortality (β = −0.50).


**Notable observations:**
Beliefs about heart failure were conditionally associated with beliefs about hypertension, but the reverse was not significant.Beliefs about atrial fibrillation and stroke were not significantly associated in either direction.Beliefs about atherosclerosis did not increase with beliefs about hypertension or atrial fibrillation, nor vice versa.Conditional associations were observed from atrial fibrillation to atherosclerosis, but not in the reverse direction.Beliefs about atherosclerosis and stroke were bidirectionally associated, suggesting consistent co-endorsement patterns among respondents.


#### 3.2.2. Metabolic Related Beliefs

Stronger beliefs about obesity co-occurred with stronger endorsement of beliefs about type 2 diabetes (β = −0.94), dyslipidemia (β = −0.37), heart failure (β = −0.37), depression (β = −0.24), kidney stones (β = −0.52), and atherosclerosis (β = −0.33). In contrast, stronger beliefs about the progression of breast cancer (β = 0.28) were linked to relatively weaker beliefs about obesity.

Stronger beliefs about dyslipidemia were found alongside stronger beliefs about obesity (β = −0.33), type 2 diabetes (β = −1.14), dementia/cognitive disorders (β = −0.42), atherosclerosis (β = −0.26), hematological disorders (β = −0.40), and progression of breast cancer (β = −0.27).

Stronger beliefs regarding type 2 diabetes tended to accompany stronger beliefs about osteoporosis (β = −0.20), obesity (β = −0.86), dyslipidemia (β = −1.17), stroke (β = −0.27), and rheumatoid arthritis (β = −0.33). However, they were comparatively weaker when beliefs about dementia/cognitive disorders were stronger (β = 0.29). Notably, beliefs of cognitive disorders/dementia were positively associated with beliefs of dyslipidemia, but inversely associated with beliefs about type 2 diabetes.

#### 3.2.3. Renal Beliefs

Stronger beliefs about chronic kidney disease tended to co-occur with stronger beliefs about heart failure (β = −0.61), cognitive disorders/dementia (β = −0.60), kidney stones (β = −0.74), hematological disorders (β = −0.29), and premature death (β = −0.32). In contrast, they were relatively weaker alongside stronger beliefs about osteoporosis (β = 0.24).

Similarly, stronger beliefs about kidney stones were observed more frequently in respondents with greater beliefs about osteoporosis (β = −0.56), obesity (β = −0.45), chronic kidney disease (β = −0.82), stomach cancer (β = −0.51), hematological disorders (β = −0.27), and rheumatological disorders (β = −0.57). However, they were comparatively weaker among those with stronger beliefs about depression (β = 0.25).


**Notable aspects:**
Beliefs about hematological disorders were positively associated with stronger beliefs about both renal-related topics.The association with osteoporosis-related beliefs showed opposing directions: positive with kidney stones, but negative with chronic kidney disease beliefs.


#### 3.2.4. Neurocognitive and Psychiatric Beliefs

Stronger beliefs about depression were more commonly observed alongside greater beliefs about osteoporosis (β = −0.34), type 2 diabetes (β = −0.24), cognitive disorders/dementia (β = −1.20), hematological disorders (β = −0.43), and progression of breast cancer (β = −0.58).

Beliefs related to cognitive disorders/dementia showed a wider range of associations, tending to co-occur with stronger beliefs about osteoporosis (β = −0.25), dyslipidemia (β = −0.42), stroke (β = −0.49), depression (β = −1.32), chronic kidney disease (β = −0.61), stomach cancer (β = −0.29), and atherosclerosis (β = −0.32). In contrast, they were comparatively weaker among those with greater beliefs about type 2 diabetes (β = 0.25).


**Notable aspects:**
Both depression and cognitive disorder/dementia beliefs increased the stronger beliefs about osteoporosis.Beliefs about type 2 diabetes showed diverging associations: co-occurrence with depression beliefs was positive, while association with beliefs about cognitive disorders/dementia were inverse.


#### 3.2.5. Rheumatologic Beliefs

Beliefs about osteoporosis tended to co-occur with stronger beliefs about atrial fibrillation (β = −0.66), type 2 diabetes (β = −0.22), depression (β = −0.42), cognitive impairment/dementia (β = −0.26), kidney stones (β = −0.58), and progression of breast cancer (β = −0.48).

Stronger beliefs regarding rheumatoid arthritis were more common alongside greater beliefs about type 2 diabetes (β = −0.30), kidney stones (β = −0.45), atherosclerosis (β = −0.37), hematological disorders (notably, β = −1.00), and progression of breast cancer (notably, β = −1.25). Notably, both osteoporosis and rheumatoid arthritis beliefs showed positive associations with beliefs about type 2 diabetes, kidney stones, and progression of breast cancer.

#### 3.2.6. Mortality Beliefs

Beliefs about premature death tended to be stronger alongside greater beliefs about hypertension (β = −0.62), atrial fibrillation (β = −0.23), dyslipidemia (β = −0.28), stroke (β = −0.47), atherosclerosis (β = −0.55), and progression of breast cancer (β = −0.33).

Premature death beliefs co-occurred with stronger beliefs about hypertension (β = −0.70), stroke (β = −0.37), chronic kidney disease (β = −0.32), and atherosclerosis (β = −0.50). This indicates that many of these associations were not reciprocal.

#### 3.2.7. Differences in Beliefs by Covariates

The analysis showed that even after adjusting for overall beliefs regarding salt intake and disease links, some demographic and individual characteristics still significantly influenced the belief structure.

Self-awareness of one’s medical history was associated with stronger beliefs about hypertension, chronic kidney disease, and rheumatoid arthritis, but showed a negative association with beliefs about dyslipidemia linked to salt intake.

Older age was related to stronger beliefs about stroke and hematological disorders but was negatively associated with beliefs about dyslipidemia, stomach cancer, and, surprisingly, premature death.

Place of residence (rural vs. urban) showed no significant association with beliefs across the studied domains, suggesting a possible equalization of health information access and a stronger role for other factors like education. Similarly, the number of health risk factors was insignificantly associated with the studied beliefs.

A notable positive association was observed between the perceived need for public health action and beliefs regarding the link between salt intake and progression of breast cancer (β = −0.79).

In summary, despite controlling for key covariates, a portion of variability in beliefs about salt-disease relationships remains unexplained, indicating the presence of other, less tangible factors influencing belief structure in the population.

### 3.3. How the Beliefs Cluster Together—Insights from the Exploratory/Confirmatory Factor Analysis (EFA/CFA)

A five-factor model (Table 4) provided an interpretable solution with acceptable fit (Tucker–Lewis Index [TLI] = 0.91; Root Mean Square Error of Approximation [RMSEA] = 0.087, 95% CI: 0.079–0.095).

Two alternative allocation rules were examined. Under the conservative rule (items retained only if their primary loading was ≥0.35 and at least 0.15 stronger than any cross-loading), three coherent multi-item domains emerged: cardiovascular beliefs (α = 0.84), metabolic beliefs (α = 0.83), and rheumatic/oncological beliefs (α = 0.82), alongside two single-item factors (cognition and kidney stones). Under the pragmatic rule, where each item was assigned to its strongest loading regardless of cross-loadings, five broader domains were obtained: cardiovascular/renal (α = 0.86), metabolic (α = 0.83), rheumatic/oncological (α = 0.82), neuro-psychological (α = 0.79), and kidney stones (single item).

The factor structure suggested by the exploratory factor analysis was further tested using confirmatory factor analysis (CFA), based on the pragmatic approach to item assignment (see Table 4 and Methods, Section 2.4). The four-factor model showed an excellent overall fit to the data (Comparative Fit Index [CFI] = 0.990; TLI = 0.988), with an acceptable level of residual error (RMSEA = 0.076, 95% CI 0.079–0.095; Standardized Root Mean Square Residual [SRMR] = 0.060).

All standardized factor loadings (λ) were high and statistically significant (all *p* < 0.001), ranging from 0.695 (perceived risk of death associated with high salt intake, loading on Factor 1) to 0.877 (perceived risk of type 2 diabetes, loading on Factor 2). This indicates that each item was strongly aligned with its corresponding latent domain. In particular, Factor 1 (cardiovascular and fatal conditions) showed consistently high loadings across eight items (λ = 0.695–0.792), while Factor 2 (metabolic conditions) and Factor 3 (non-cardiometabolic chronic diseases) both showed strong internal coherence (λ typically >0.74). Factor 4 (cognitive and depressive disorders) also demonstrated very strong and balanced loadings (both λ = 0.849).

Modification indices (MIs) were inspected to evaluate potential cross-loadings or residual correlations (Table 5, lower panel). Although several MIs indicated that the model fit could be marginally improved by allowing additional cross-loadings (e.g., osteoporosis or stomach cancer loading on multiple factors), we did not implement these modifications to avoid overfitting and to maintain conceptual clarity of the factor structure.

Taken together, the CFA results confirm that the beliefs about health risks associated with high salt consumption can be meaningfully represented by four correlated latent domains, supporting the factor structure derived from the exploratory analysis. Figure 2 presents a bar-style plot of standardized factor loadings.

## 4. Discussion

Several studies have addressed common misconceptions regarding salt intake and its influence on dietary behavior. For instance, Sarmugam and Ball demonstrated that salt-related knowledge and taste-related beliefs significantly affect the use of discretionary salt, highlighting the mediating role of false assumptions in salt consumption patterns [32]. Likewise, Morowatisharifabad et al. identified a range of prevalent myths, such as the belief that salt reduces blood lipids or improves digestion, which serve as barriers to behavioral change in at-risk populations [33]. Recent critical reviews have also challenged misleading claims suggesting that low salt intake may be harmful; Cappuccio emphasized that such claims often stem from methodological flaws and commercial interests rather than reliable evidence [34]. Furthermore, WHO-affiliated researchers have highlighted serious limitations in studies supporting a J-shaped association between sodium intake and cardiovascular outcomes, often due to inadequate sodium measurement methods [35]. Building on this research, the present study introduces a novel belief-based analytical framework that examines how individual beliefs interact within structured systems. Importantly, beyond these conditional associations, factor-analytic evidence confirmed that the reported items do not function as isolated measures but reflect broader, internally consistent latent constructs. By applying multivariate POLR model [23,24,36], this approach allows interpretation of how specific beliefs, conditionally, influence attitudes and behaviors, offering new insights into the cognitive mechanisms shaping salt-related health literacy. So, our study employed a multivariate analytical strategy based on POLR models to explore the structure of public beliefs regarding the health consequences of excessive salt intake. Unlike traditional psychometric methods, this approach treats beliefs not as isolated items but as interconnected elements within a broader cognitive system. The analysis accounted for demographic and psychosocial factors and revealed direct and asymmetrical links between belief domains. This modeling suggests cognitive clusters and hierarchies, providing a system-level view of salt-related health literacy. These asymmetries were not only visible in pairwise POLR associations, but also mirrored in the factor structure, where cardiovascular items clustered tightly while remaining separable from metabolic or neurocognitive domains. This interpretation is further supported by the results of our factor analyses, which demonstrated that individual beliefs indeed clustered into broader latent domains (e.g., cardiovascular, metabolic, neuro-psychological). This confirms that the observed pairwise associations are embedded within higher-order structures rather than being random co-occurrences.

Recent studies show nutrition literacy is about integrating factual knowledge into coherent belief systems [37]. Our study extends this view by demonstrating that beliefs about salt-related disease risks are organized into quasi-hierarchical patterns, supporting a more dynamic model of health cognition [38,39].

Asymmetries likely reflect heuristics—people focus on well-known links like salt and hypertension, ignoring less known risks like cognitive decline or chronic kidney disease [25]. This pattern highlights the need for balanced messaging across disease domains.

We identified key concepts like hypertension and mortality risk that connect broader salt-related beliefs and could guide education. Public health campaigns anchored in these central salt-related beliefs may achieve wider conceptual generalization and behavioral impact.

Even medically linked respondents showed fragmented beliefs; for example, atrial fibrillation was not connected to stroke, revealing gaps that might hinder prevention.

Future research should test if reshaping belief networks via digital tools improves diet and risk awareness; we found no causal proof yet. Experimental interventions that directly address the architecture of belief may represent a promising new direction in nutrition education.

Beliefs about certain diseases cluster meaningfully, for example, hypertension links strongly to stroke, heart failure, and kidney disease. Similarly, beliefs about metabolic diseases such as obesity, type 2 diabetes, and dyslipidemia formed a tightly integrated network, suggesting a shared public mental model of these conditions as diet-related and mutually reinforcing.

Some links were asymmetric: knowing heart failure relates to hypertension, but not the reverse; atrial fibrillation raised beliefs about atherosclerosis, but not vice versa. These asymmetries may indicate conceptual hierarchies, but direction does not prove causality. This notion aligns with the work of van der Linden et al., who demonstrated that modifying a central belief in a cognitive network can propagate changes across the system [40]. In our study, beliefs regarding mortality and hypertension emerged as possible cognitive hubs—key anchoring points within the salt-related belief network that may represent central nodes within the belief network, potentially influencing how information is integrated, though causal effects remain unconfirmed. Our factor-analytic (EFA/CFA) results are consistent with this notion, as mortality and hypertension emerged not only as central nodes in the conditional association models but also loaded strongly on the same latent factor, reinforcing their role as cognitive hubs.

These patterns also resonate with recent advances in network-based health communication theory. Schwarz and Jalbert argue that targeting central nodes in a belief structure—those most interconnected with others—can (hypothetically) amplify the effectiveness of public health messaging [41]. The prominence of mortality and hypertension in our findings supports their role as optimal intervention targets. Rather than disseminating isolated facts, future campaigns could focus on reinforcing such hubs to maximize cognitive reach.

The cognitive linkage between depression, osteoporosis, and dementia is also noteworthy. These domains, though traditionally seen as less directly related to dietary sodium, were positively intercorrelated, implying either shared health messaging (e.g., aging-related vulnerability) or public heuristics associating “unhealthy diet” with general decline. This pattern further supports the idea that salt-related belief systems are shaped not only by informational accuracy, but also by affective and intuitive processes. Slovic et al. showed that people often rely on emotional resonance rather than objective risk probabilities when evaluating health threats [42]. In our data, the strong salience of conditions like stroke and death—both emotionally charged and highly visible—suggests that affective cues may contribute to their centrality in salt-related reasoning, while more abstract outcomes like cancer remain cognitively distant. This interconnection was also visible in the factor structure, where depression and cognitive decline consistently clustered, suggesting that laypeople may perceive them as a shared vulnerability domain rather than distinct conditions.

Moreover, our findings on cognitive fragmentation highlight gaps in lay reasoning. The absence of associations between atrial fibrillation and stroke, or between hypertension and heart failure, despite their clear biomedical links, suggests that lay belief networks do not always mirror clinical reality. Interestingly, while these gaps were evident in the conditional network, the factor analyses suggested that such items still shared variance within broader latent domains, highlighting a tension between item-level fragmentation and domain-level coherence. This suggests that local disjunctions between specific items do not necessarily undermine coherence at the broader domain level. Ahn et al. noted that people tend to favor proximate, mechanistic explanations over distal or probabilistic ones when reasoning about causality [43]. This may explain the conceptual isolation of diseases like breast or stomach cancer in relation to salt, which require more abstract causal understanding.

The results align with Bruner’s constructivism: factual knowledge is actively built through meaning connections, not passively received [44]. In our case, beliefs about salt and disease risks appear to function within an emergent cognitive architecture—structured by both logic and experience—rather than as a list of discrete facts. Health literacy should thus be viewed not just as correctness of factual understanding, but as the coherence and connectivity of belief structures [45].

Demographics like self-awareness affected beliefs, but factors like residence had little impact, hinting that access is not the same as quality. As Chou et al. observed, digital and social media environments can create echo chambers that reinforce fragmented or inaccurate beliefs, even among educated populations. The web-based format of our study makes this particularly relevant: exposure to information does not guarantee conceptual integration [46].

This study goes beyond classic psychometrics to view salt-related beliefs as a structured cognitive network. Using a multivariate, conditionally adjusted approach, we demonstrate that health-related beliefs form non-trivial, asymmetrical associations that suggest latent reasoning pathways—not merely factual knowledge or ignorance.

Certain domains, particularly cardiovascular and metabolic conditions, show internal coherence and mutual reinforcement, implying intuitive or culturally mediated cognitive schemata. Others, such as neurocognitive disorders or mortality risks, appear to serve as bridging constructs or hubs, anchoring otherwise disparate elements of belief. Importantly, our findings reveal areas of cognitive fragmentation—such as the conceptual disjunction between hypertension and heart failure—that may reflect genuine gaps in public understanding.

Our multivariate analysis reveals a structured, interconnected architecture of public beliefs about salt-related health risks. Cardiovascular and metabolic diseases cluster tightly, reflecting intuitive cognitive schemata, while neurocognitive and psychiatric domains form overlapping but distinct groups. Importantly, asymmetrical associations—such as from heart failure to hypertension but not vice versa—suggest latent conceptual hierarchies rather than simple co-occurrence.

These findings indicate not just gaps in factual understanding but fragmentation of public belief networks, exemplified by the unexpected disconnection between atrial fibrillation and stroke, despite clinical links. Emotional salience and heuristic biases may underpin the prominence of mortality and hypertension beliefs, pointing to potential leverage points for targeted health communication. While demographic factors influence belief patterns, substantial unexplained variance suggests deeper cognitive and sociocultural drivers at play. Future research should experimentally test interventions aimed at restructuring these belief networks to improve dietary behavior.

Covariate effects, especially self-awareness of personal health parameters, accounted for meaningful variance but failed to exhaustively explain belief patterns. This suggests that other, less tangible factors may contribute to belief patterns.

### 4.1. Practical Implications

Factor-analytic clustering indicates that public reasoning operates in broader domains (e.g., cardiovascular vs. metabolic). Educational tools should therefore not only target isolated beliefs but explicitly frame them within these broader domains to reinforce internal coherence.Health education should emphasize the interdependencies between disease risks, not just isolated facts about salt intake.Public health campaigns may benefit from addressing ‘gateway beliefs’ like mortality or hypertension to reinforce broader belief networks.Cognitive fragmentation (e.g., separating heart failure from hypertension) should be targeted with tailored messaging.The asymmetry in belief associations suggests people update knowledge selectively—interventions should account for directional learning.Digital tools or infographics can visualize these belief networks for patient education or health professional training.

### 4.2. Strengths and Limitations of the Study

This study offers several notable strengths contributing to its methodological and conceptual robustness. First, by modeling the structure of beliefs rather than merely their magnitude, the analytical approach captures how accurate beliefs and misconceptions co-occur and interrelate, offering a system-level view of public understanding. This moves beyond the treatment of items as isolated units and instead reflects the interdependencies within individuals’ health-related belief systems. The addition of EFA and CFA further strengthened methodological rigor, as it allowed us to test whether the observed associations form stable latent structures. A further strength is the convergence between item-level (POLR) and domain-level (EFA/CFA) analyses, which mutually reinforce the validity of the observed structures. However, as factor solutions can be sample-specific, replication in independent cohorts remains necessary. The excellent CFA fit indices support the validity of our measurement model. Nonetheless, factor solutions may be sample-specific, and replication in independent datasets would strengthen external validity.

Secondly, the use of POLR enables ordinal modeling of beliefs, aligning with the ordered nature of the composite index—from harmful misconceptions to accurate understanding. This constitutes an advancement over traditional techniques such as principal component analysis or binary logistic regression, which typically fail to address the ordinal structure of health literacy data.

The model supports the examination of directional associations among belief elements. The beta coefficient matrix provides insight into which beliefs are statistically associated with others, revealing potential hierarchical or clustering tendencies in how individuals organize health-related beliefs. While all associations remain observational and non-causal, observed asymmetries (e.g., beliefs about condition A being associated with beliefs of condition B, but not vice versa) suggest the presence of structured, non-reciprocal patterns in conceptual organization, consistent with theoretical models of cognitive architecture in health literacy research.

Furthermore, the modeling strategy is well-suited to the nature of the survey data and the exploratory aims of the study. The method avoids assumptions of linearity or homoscedasticity without compromising analytical integrity, making it appropriate for this type of belief-structure analysis.

There were several limitations of this study. First and foremost, the questionnaire was developed by the authors and, while based on existing literature and expert input, has not undergone formal psychometric validation. This may impact the construct validity of certain measures and limits the generalizability of the results.

Moreover, the absence of conventional model assumption diagnostics—though methodologically defensible given the data structure—may nonetheless be viewed critically by audiences more accustomed to classical regression frameworks. Moreover, the use of pairwise associations, rather than more comprehensive modeling frameworks such as Bayesian networks or structural equation modeling (SEM), restricts the ability to account for indirect effects or latent common factors. This constrains the interpretability of more complex interdependencies within the belief system.

Additionally, one item in the belief index—assessing the perceived link between salt intake and risk of death—was designed without a neutral midpoint. While the absence of a neutral option was originally intended to mitigate central tendency bias in response to emotionally salient outcomes, this decision was not applied consistently across all items and was not supported by an a priori justification in the initial design protocol. As such, we acknowledge that this inconsistency in response format may have influenced respondents’ likelihood of expressing directional opinions on this item. Although the proportional odds modeling approach remains robust to such variations, the lack of a neutral midpoint may have subtly shaped response distributions and should be addressed in future survey designs for improved comparability.

Post hoc analyses were not conducted for between-group univariate comparisons because the primary analytical focus was on multivariate proportional odds logistic regression models, which included occupational association with medicine as a covariate. This approach allowed for adjustment of group effects within a unified modeling framework, rendering separate post hoc tests unnecessary.

Interpretation also requires caution: for example, negative coefficients may reflect seemingly counterintuitive associations, and the directionality of associations should not be interpreted as indicative of temporal or causal influence. Finally, some redundancy among belief items—stemming from overlapping content or similar public messaging—could obscure deeper conceptual distinctions, as some observed associations may result from surface-level semantic similarity rather than meaningful cognitive integration.

## 5. Conclusions

Belief systems regarding salt-related disease risks appear to be structured, multifaceted, and only partially aligned with biomedical consensus. The identified latent-domain structure confirms that these beliefs are not random, but systematically clustered, reflecting higher-order cognitive schemata. This dual evidence—from both conditional associations and latent-domain clustering—underscores that effective interventions must address not only individual misconceptions but also the higher-order belief domains in which they are embedded. Our findings indicate that public understanding is not a simple sum of discrete facts, but a network of beliefs that may be internally coherent yet diverge from clinical frameworks.

These results suggest that public health efforts should address not only factual inaccuracies, but also the configuration of belief systems that shape how new information is processed, integrated, or resisted. The persistence of misconceptions may reflect more than informational gaps; they may emerge from how individuals internally relate different health concepts.

Rather than aiming solely to correct isolated errors, health communication strategies may benefit from engaging with the underlying structure of beliefs. By identifying domains where conceptual fragmentation or selective associations are most apparent, such strategies can be tailored to the cognitive tendencies of the audience.

While our analyses identify patterns of association that resemble hierarchical or clustered reasoning pathways, it is crucial to emphasize that these structures are not evidence of causal mechanisms. They reflect statistical relationships within reported beliefs—patterns that may arise from shared messaging, heuristics, emotional salience, or cognitive shortcuts, rather than from logical or educational progression.

Ultimately, the architecture of public belief appears to be both patterned and imperfect—shaped by experience, exposure, intuition, and selective attention. Recognizing this complexity is essential for designing educational interventions that move beyond the correction of isolated facts toward more integrative, psychologically informed approaches to improving health literacy.

## Figures and Tables

**Figure 1 nutrients-17-02728-f001:**
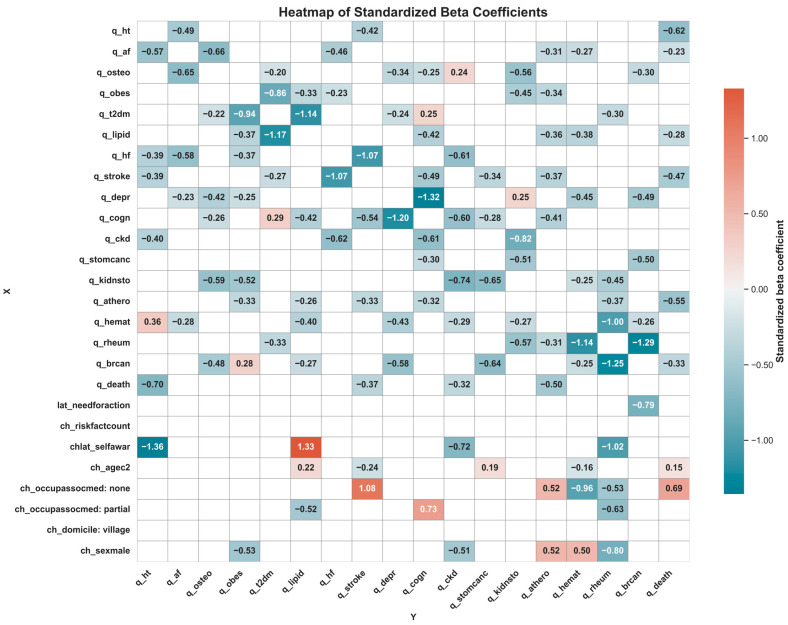
Heatmap of standardized beta coefficients from regression models showing associations between predictors (*X*-axis) and outcome variables (*Y*-axis). Each cell displays the strength and direction of the effect of a predictor on an outcome, with red shades indicating negative effects and blue shades positive effects. Cells left blank represent no statistically significant association. **Variable abbreviations and meaning:** q_saltcauseht—belief that salt causes hypertension; q_saltcauseendoth—belief that salt causes endothelial damage; q_saltcauseinflam—belief that salt induces inflammation; q_saltcausefatdysf—belief that salt causes adipose tissue dysfunction; q_saltcausenone—belief that salt causes none of the listed mechanisms; q_educat—belief in the need for public education on salt; q_infoverif—belief in the need to verify existing information about salt; q_legisl—belief in the need for legislative action on salt; q_noaction—belief that no action regarding education, verification, or legislation is needed; q_ht—belief that excessive salt contributes to hypertension; q_af—belief that excessive salt contributes to atrial fibrillation; q_osteo—belief that excessive salt contributes to osteoporosis; q_obes—belief that excessive salt contributes to obesity; q_t2dm—belief that excessive salt contributes to type 2 diabetes; q_lipid—belief that excessive salt contributes to dyslipidemia; q_hf—belief that excessive salt contributes to heart failure; q_stroke—belief that excessive salt contributes to stroke; q_depr—belief that excessive salt contributes to depression; q_cogn—belief that excessive salt contributes to cognitive decline or dementia; q_ckd—belief that excessive salt contributes to chronic kidney disease; q_stomcanc—belief that excessive salt contributes to stomach cancer; q_kidnsto—belief that excessive salt contributes to kidney stones; q_athero—belief that excessive salt contributes to atherosclerosis; q_hemat—belief that excessive salt contributes to hematologic disorders; q_rheum—belief that excessive salt contributes to rheumatoid arthritis; q_brcan—belief that excessive salt influences breast cancer progression; q_death—belief that excessive salt increases the risk of premature death; lat_needforaction—perceived general need for public health interventions related to salt; chlat_selfawar—verified self-awareness of cardiovascular and metabolic health indicators (e.g., SBP, DBP, lipids, glucose); ch_agec2—age category, centered at 41–50 years; ch_sexmale—male sex; ch_domicile—place of residence; ch_occupassocmed—occupational affiliation with medicine; ch_riskfactcount—number of self-declared cardiometabolic risk factors.

**Figure 2 nutrients-17-02728-f002:**
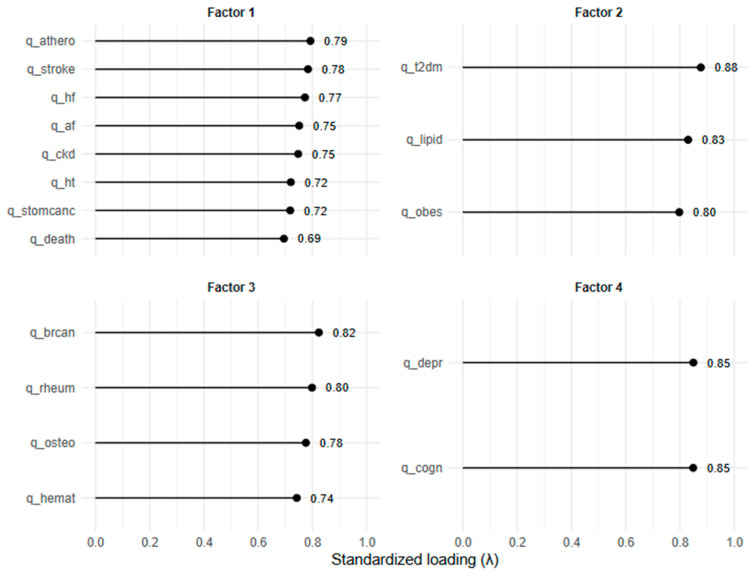
Standardized factor loadings from the confirmatory factor analysis (CFA), visualized as lollipop plots. Each panel corresponds to one latent factor (“Factor 1–4”), with items on the vertical axis and their standardized loadings (λ) on the horizontal axis. Numerical values next to the points indicate the exact standardized loadings.

**Table 1 nutrients-17-02728-t001:** Population sample characteristics after stratification in terms of occupational medical background. The meaning of the belief aspects (column ‘Features’) is shown in the ‘Variable overview’ section.

Feature	MDs (N = 321)		Medical, Not MDs (N = 157)		Non-Medical (N = 190)		Levene *p*	*p*
	µ	SD	µ	SD	µ	SD		
q_saltcauseht	0.97	0.17	0.90	0.30	0.85	0.36	<0.001	<0.001
q_saltcauseendoth	0.68	0.47	0.50	0.50	0.23	0.42	<0.001	<0.001
q_saltcauseinflam	0.58	0.49	0.49	0.50	0.21	0.40	<0.001	<0.001
q_saltcausefatdysf	0.38	0.49	0.35	0.48	0.26	0.44	<0.001	0.024
q_saltcausenone	0.02	0.15	0.05	0.22	0.19	0.40	<0.001	<0.001
q_educat	0.97	0.17	0.98	0.14	0.97	0.18	0.339	0.765
q_infoverif	0.41	0.49	0.50	0.50	0.38	0.49	0.016	0.065
q_noaction	0.03	0.17	0.04	0.19	0.02	0.12	0.030	0.339
q_legisl	0.49	0.50	0.50	0.50	0.36	0.48	<0.001	0.005
q_ht	1.82	0.56	1.73	0.70	1.53	0.72	<0.001	<0.001
q_af	0.97	0.95	0.83	1.10	0.65	0.99	0.007	0.002
q_osteo	0.33	1.11	0.43	1.19	0.27	1.02	0.024	0.396
q_obes	0.88	1.13	1.01	1.14	0.91	1.04	0.331	0.523
q_t2dm	0.47	1.17	0.57	1.26	0.45	1.07	0.008	0.597
q_lipid	0.47	1.11	0.80	1.15	0.67	0.98	0.016	0.006
q_hf	1.51	0.75	1.35	0.93	1.07	0.91	0.098	<0.001
q_stroke	1.51	0.75	1.27	0.97	0.79	0.97	0.001	<0.001
q_depr	0.02	0.99	0.02	1.03	−0.12	1.02	0.343	0.273
q_cogn	0.53	1.11	0.35	1.13	0.21	1.04	0.156	0.005
q_ckd	1.42	0.81	1.43	0.93	1.27	0.85	0.695	0.125
q_stomcanc	0.65	1.12	0.69	1.16	0.55	1.02	0.060	0.431
q_kidnsto	0.79	1.03	1.03	1.05	0.93	0.96	0.190	0.051
q_athero	1.24	0.91	1.18	1.01	0.92	1.00	0.334	0.001
q_hemat	−0.06	0.99	0.29	1.07	0.41	0.95	0.008	<0.001
q_rheum	0.12	1.03	0.52	1.09	0.32	0.94	0.033	0.001
q_brcan	−0.05	0.96	0.13	1.06	−0.09	0.88	0.009	0.099
q_death	1.55	0.69	1.44	0.84	0.99	0.99	0.216	<0.001

MDs stands for Medical Doctors. Columns: µ, mean value; SD, standard deviation; Levene *p*, *p*-value from the Levene’s test for homoscedasticity (equality of variance between the groups); *p*, *p*-value for the ANOVA (or Welch ANOVA in case of heteroscedasticity). Variable abbreviations and meaning: q_saltcauseht—belief that salt causes hypertension; q_saltcauseendoth—belief that salt causes endothelial damage; q_saltcauseinflam—belief that salt induces inflammation; q_saltcausefatdysf—belief that salt causes adipose tissue dysfunction; q_saltcausenone—belief that salt causes none of the listed mechanisms; q_educat—belief in the need for public education on salt; q_infoverif—belief in the need to verify existing information about salt; q_legisl—belief in the need for legislative action on salt; q_noaction—belief that no action regarding education, verification, or legislation is needed; q_ht—belief that excessive salt contributes to hypertension; q_af—belief that excessive salt contributes to atrial fibrillation; q_osteo—belief that excessive salt contributes to osteoporosis; q_obes—belief that excessive salt contributes to obesity; q_t2dm—belief that excessive salt contributes to type 2 diabetes; q_lipid—belief that excessive salt contributes to dyslipidemia; q_hf—belief that excessive salt contributes to heart failure; q_stroke—belief that excessive salt contributes to stroke; q_depr—belief that excessive salt contributes to depression; q_cogn—belief that excessive salt contributes to cognitive decline or dementia; q_ckd—belief that excessive salt contributes to chronic kidney disease; q_stomcanc—belief that excessive salt contributes to stomach cancer; q_kidnsto—belief that excessive salt contributes to kidney stones; q_athero—belief that excessive salt contributes to atherosclerosis; q_hemat—belief that excessive salt contributes to hematologic disorders; q_rheum—belief that excessive salt contributes to rheumatoid arthritis; q_brcan—belief that excessive salt influences breast cancer progression; q_death—belief that excessive salt increases the risk of premature death; lat_needforaction—perceived general need for public health interventions related to salt; chlat_selfawar—verified self-awareness of one’s cardiovascular and metabolic health indicators (e.g., SBP, DBP, lipids, glucose); ch_agec2—age category, centered at 41–50 years; ch_sexmale—sex (male = 1, female = 0); ch_domicile—place of residence (rural = 1, urban = 0); ch_occupassocmed—occupational affiliation with medicine (0 = none, 1 = allied health, 2 = physician); ch_riskfactcount—number of self-declared cardiometabolic risk factors.

**Table 2 nutrients-17-02728-t002:** The full map of standardized beta-coefficients based on modeling each belief aspect (columns). Statistical significance was assessed at α = 0.05. Exact *p*-values are provided in Table 3. The structure reflects adjusted, directional conditional associations under multivariate control. Covariates appear below the horizontal division.

Feature	q_ht	q_af	q_osteo	q_obes	q_t2dm	q_lipid	q_hf	q_stroke	q_depr	q_cogn	q_ckd	q_stomcanc	q_kidnsto	q_athero	q_hemat	q_rheum	q_brcan	q_death
q_ht		−0.49	0.02	−0.10	−0.09	−0.02	−0.25	−0.42	−0.07	0.17	−0.29	−0.14	0.02	0.05	0.11	0.07	−0.09	−0.62
q_af	−0.57		−0.66	−0.05	−0.10	−0.03	−0.46	−0.13	−0.16	0.04	−0.09	−0.16	0.03	−0.31	−0.27	0.06	0.03	−0.23
q_osteo	0.06	−0.65		−0.11	−0.20	0.03	−0.06	0.01	−0.34	−0.25	0.24	−0.16	−0.56	−0.05	0.09	−0.01	−0.30	0.11
q_obes	−0.23	−0.02	−0.07		−0.86	−0.33	−0.23	−0.03	−0.18	0.11	−0.12	0.09	−0.45	−0.34	0.02	0.17	0.19	−0.09
q_t2dm	0.07	−0.12	−0.22	−0.94		−1.14	−0.09	−0.21	−0.24	0.25	−0.17	−0.17	0.16	0.06	−0.12	−0.30	−0.01	0.18
q_lipid	−0.08	0.05	0.05	−0.37	−1.17		−0.23	0.09	0.07	−0.42	0.09	0.16	−0.02	−0.36	−0.38	0.15	−0.18	−0.28
q_hf	−0.39	−0.58	−0.11	−0.37	−0.10	−0.21		−1.07	−0.12	0.23	−0.61	−0.06	0.01	0.04	0.08	−0.12	−0.03	−0.12
q_stroke	−0.39	−0.23	−0.02	−0.01	−0.27	0.03	−1.07		−0.09	−0.49	−0.09	−0.34	0.03	−0.37	0.20	0.06	0.14	−0.47
q_depr	−0.13	−0.23	−0.42	−0.25	−0.19	0.05	−0.04	−0.03		−1.32	0.24	0.04	0.25	0.23	−0.45	−0.16	−0.49	−0.02
q_cogn	0.20	0.07	−0.26	0.04	0.29	−0.42	0.17	−0.54	−1.20		−0.60	−0.28	−0.04	−0.41	−0.06	−0.02	−0.13	0.02
q_ckd	−0.40	−0.06	0.22	−0.06	−0.19	0.04	−0.62	−0.14	0.24	−0.61		−0.19	−0.82	−0.12	−0.23	0.07	0.05	−0.24
q_stomcanc	−0.13	−0.14	−0.17	0.06	−0.14	0.12	−0.02	−0.20	0.03	−0.30	−0.16		−0.51	−0.05	−0.13	0.03	−0.50	−0.11
q_kidnsto	0.01	0.06	−0.59	−0.52	0.18	−0.05	0.06	−0.01	0.17	0.01	−0.74	−0.65		−0.21	−0.25	−0.45	0.05	0.02
q_athero	−0.11	−0.20	−0.13	−0.33	0.14	−0.26	0.06	−0.33	0.20	−0.32	−0.11	0.00	−0.16		−0.15	−0.37	−0.03	−0.55
q_hemat	0.36	−0.28	0.09	−0.01	−0.18	−0.40	0.10	0.18	−0.43	−0.10	−0.29	−0.15	−0.27	−0.17		−1.00	−0.26	0.23
q_rheum	0.19	0.11	0.00	0.21	−0.33	0.15	−0.07	0.10	−0.18	−0.05	0.13	0.14	−0.57	−0.31	−1.14		−1.29	−0.01
q_brcan	−0.05	−0.02	−0.48	0.28	−0.11	−0.27	−0.05	0.00	−0.58	−0.15	0.09	−0.64	−0.02	−0.03	−0.25	−1.25		−0.33
q_death	−0.70	−0.08	0.07	−0.09	0.21	−0.19	−0.16	−0.37	0.06	0.13	−0.32	−0.11	0.16	−0.50	0.17	−0.05	−0.21	
lat_needforaction	−0.33	−0.49	0.31	0.06	−0.25	0.45	0.44	0.02	0.03	−0.36	0.18	−0.10	−0.19	−0.03	0.28	0.35	−0.79	−0.53
ch_riskfactcount	0.01	−0.09	0.07	−0.07	−0.02	0.00	−0.01	−0.03	0.00	−0.07	−0.08	0.04	0.09	0.02	0.06	−0.06	0.03	0.05
chlat_selfawar	−1.36	0.20	−0.03	−0.38	0.06	1.33	−0.33	0.01	0.08	−0.03	−0.72	0.13	0.39	0.31	0.47	−1.02	0.13	−0.24
ch_agec2	−0.12	0.07	−0.03	0.08	−0.10	0.22	0.07	−0.24	0.01	0.00	0.00	0.19	0.07	−0.01	−0.16	0.00	−0.01	0.15
ch_occupassocmed: none	0.07	0.21	−0.25	−0.41	0.39	−0.31	0.43	1.08	0.14	0.40	−0.21	−0.20	−0.20	0.52	−0.96	−0.53	0.18	0.69
ch_occupassocmed: partial	−0.26	0.33	−0.27	−0.21	0.35	−0.52	0.25	0.16	0.08	0.73	−0.45	0.01	−0.06	0.14	−0.39	−0.63	0.06	0.16
ch_domicile: village	0.01	0.04	0.11	−0.30	0.09	−0.02	0.20	0.03	−0.02	−0.25	0.55	0.22	−0.48	0.22	0.01	−0.26	0.09	−0.34
ch_sexmale	−0.40	0.00	−0.01	−0.53	−0.02	−0.10	0.16	0.25	0.23	−0.15	−0.51	−0.03	−0.27	0.52	0.50	−0.80	0.16	0.27

The red color marks statistically-significant (*p* < 0.05) beta-coefficients.

**Table 3 nutrients-17-02728-t003:** The full map of *p*-values associated with each beta-coefficient based on modeling each belief aspect (columns). This table is auxiliary to Table 2, where the standardized beta-coefficients are displayed.

Feature	q_ht	q_af	q_osteo	q_obes	q_t2dm	q_lipid	q_hf	q_stroke	q_depr	q_cogn	q_ckd	q_stomcanc	q_kidnsto	q_athero	q_hemat	q_rheum	q_brcan	q_death
q_ht		0.002	0.922	0.522	0.575	0.909	0.116	0.009	0.673	0.294	0.074	0.378	0.875	0.730	0.495	0.661	0.586	<0.001
q_af	<0.001		<0.001	0.657	0.331	0.739	<0.001	0.244	0.122	0.669	0.435	0.108	0.770	0.004	0.011	0.560	0.753	0.048
q_osteo	0.678	<0.001		0.256	0.044	0.752	0.571	0.921	<0.001	0.010	0.039	0.092	<0.001	0.647	0.345	0.917	0.002	0.332
q_obes	0.101	0.840	0.491		<0.001	0.001	0.030	0.747	0.074	0.260	0.256	0.338	<0.001	0.001	0.842	0.096	0.071	0.406
q_t2dm	0.659	0.245	0.027	<0.001		<0.001	0.450	0.062	0.020	0.014	0.136	0.099	0.119	0.601	0.241	0.004	0.919	0.116
q_lipid	0.623	0.640	0.664	0.001	<0.001		0.057	0.436	0.550	<0.001	0.456	0.118	0.878	0.001	0.001	0.159	0.112	0.019
q_hf	0.015	<0.001	0.390	0.004	0.434	0.103		<0.001	0.370	0.073	<0.001	0.622	0.962	0.761	0.542	0.343	0.819	0.388
q_stroke	0.017	0.067	0.881	0.951	0.034	0.795	<0.001		0.501	<0.001	0.517	0.005	0.811	0.003	0.126	0.657	0.289	0.001
q_depr	0.446	0.047	<0.001	0.037	0.090	0.690	0.768	0.815		<0.001	0.081	0.727	0.033	0.054	<0.001	0.174	<0.001	0.901
q_cogn	0.234	0.482	0.011	0.732	0.005	<0.001	0.151	<0.001	<0.001		<0.001	0.006	0.731	<0.001	0.553	0.847	0.226	0.875
q_ckd	0.015	0.653	0.074	0.616	0.125	0.770	<0.001	0.298	0.062	<0.001		0.123	<0.001	0.329	0.080	0.591	0.704	0.079
q_stomcanc	0.370	0.114	0.064	0.497	0.129	0.209	0.848	0.054	0.785	0.001	0.126		<0.001	0.619	0.166	0.737	<0.001	0.284
q_kidnsto	0.959	0.604	<0.001	<0.001	0.108	0.634	0.643	0.902	0.131	0.942	<0.001	<0.001		0.052	0.024	<0.001	0.644	0.900
q_athero	0.475	0.071	0.240	0.003	0.212	0.019	0.617	0.005	0.072	0.004	0.382	0.997	0.162		0.196	0.001	0.814	<0.001
q_hemat	0.028	0.012	0.398	0.927	0.110	<0.001	0.433	0.145	<0.001	0.376	0.022	0.176	0.019	0.152		<0.001	0.021	0.074
q_rheum	0.290	0.353	0.984	0.089	0.005	0.219	0.623	0.445	0.133	0.657	0.327	0.215	<0.001	0.012	<0.001		<0.001	0.951
q_brcan	0.773	0.841	<0.001	0.027	0.363	0.029	0.745	0.977	<0.001	0.228	0.510	<0.001	0.860	0.831	0.040	<0.001		0.018
q_death	<0.001	0.501	0.566	0.448	0.085	0.105	0.190	0.002	0.644	0.280	0.013	0.348	0.194	<0.001	0.167	0.705	0.088	
lat_needforaction	0.460	0.093	0.273	0.845	0.391	0.130	0.184	0.940	0.921	0.215	0.587	0.723	0.536	0.920	0.346	0.242	0.008	0.109
ch_riskfactcount	0.913	0.055	0.167	0.158	0.738	0.942	0.923	0.580	0.952	0.176	0.161	0.343	0.082	0.765	0.186	0.239	0.535	0.410
chlat_selfawar	0.004	0.509	0.917	0.222	0.847	<0.001	0.351	0.972	0.800	0.921	0.042	0.661	0.219	0.344	0.130	0.001	0.683	0.495
ch_agec2	0.215	0.282	0.586	0.229	0.115	0.001	0.377	0.001	0.904	0.978	0.973	0.002	0.286	0.911	0.011	0.963	0.925	0.046
ch_occupassocmed: none	0.825	0.384	0.283	0.094	0.103	0.196	0.107	<0.001	0.560	0.100	0.444	0.380	0.418	0.039	<0.001	0.031	0.454	0.010
ch_occupassocmed: partial	0.453	0.128	0.198	0.358	0.109	0.019	0.325	0.518	0.724	0.001	0.082	0.964	0.788	0.540	0.072	0.005	0.788	0.511
ch_domicilevillage	0.973	0.859	0.656	0.219	0.704	0.940	0.473	0.908	0.937	0.295	0.056	0.364	0.051	0.394	0.952	0.286	0.716	0.207
ch_sexmale	0.112	0.987	0.969	0.003	0.923	0.571	0.407	0.203	0.184	0.371	0.010	0.841	0.122	0.004	0.004	<0.001	0.370	0.174

The red color marks *p*-values below the 0.05 cut-off.

**Table 4 nutrients-17-02728-t004:** The clusters formed by beliefs about high salt consumption association with selected diseases, according to the Exploratory Factor Analysis (EFA), with additional insights on factor internal consistency and model diagnostics.

Factor Loadings					
Item (Variable)	Factor 1	Factor 2	Factor 3	Factor 4	Factor 5
q_ht	0.866				
q_af	0.522				
q_hf	0.700				
q_stroke	0.738				
q_death	0.724				
q_obes		0.720			
q_t2dm		0.863			
q_lipid		0.743			
q_hemat			0.504		
q_rheum			0.691		
q_brcan			0.773		
q_cogn				0.986	
q_kidnsto					0.961
q_osteo					
q_depr			0.356	0.452	
q_ckd	0.400				0.359
q_stomcanc	0.304				
q_athero	0.372				
Internal consistency—according to the conservative approach					
Factor	Items			Number of items	Cronbach’s α
1	q_ht, q_af, q_hf, q_stroke, q_athero, q_death			6	0.841
2	q_obes, q_t2dm, q_lipid			3	0.829
3	q_hemat, q_rheum, q_brcan			3	0.824
4	q_cogn			1	-
5	q_kidnsto			1	-
Internal consistency—according to the pragmatic approach					
Factor	Items			Number of items	Cronbach’s α
1	q_th, q_af, q_hf, q_stroke, q_ckd, q_stomcanc, q_athero, q_death			8	0.862
2	q_obes, q_t2dm, q_lipid			3	0.829
3	q_osteo, q_hemat, q_rheum, q_brcan			4	0.819
4	q_depr, q_cogn			2	0.792
5	q_kidnsto			1	-
Diagnostics					
TLI = 0.9063, RMSEA: 0.0867 (95% CI: 0.0791–0.0947, confidence = 0.900)					
	Factor 1	Factor 2	Factor 3	Factor 4	Factor 5
SS loadings	3.086	2.055	1.702	1.327	1.243
Proportion variance	0.171	0.114	0.095	0.074	0.069
Cumulative variance	0.171	0.286	0.380	0.454	0.523

Factor loadings are standardized regression weights from the exploratory factor analysis (EFA), indicating the strength of association between each belief item and the corresponding latent factor. Factor extraction and rotation were performed as described in Section 2.4. Item clustering was evaluated according to two assignment rules: a conservative approach (items assigned only if the primary loading was ≥0.35 and ≥0.15 stronger than any cross-loading) and a pragmatic approach (items assigned to the factor with the highest loading, even if cross-loadings were present). Internal consistency of each factor was assessed with Cronbach’s α (values ≥ 0.70 generally indicate acceptable reliability). Model diagnostics include the Tucker–Lewis Index (TLI) and Root Mean Square Error of Approximation (RMSEA, with 95% confidence interval). SS loadings represent eigenvalues of the extracted factors; “Proportion variance” indicates the fraction of variance explained by each factor; “Cumulative variance” is the cumulative proportion explained by the solution.

**Table 5 nutrients-17-02728-t005:** Confirming the structure of clusters (factors) formed by beliefs about high salt consumption association with selected diseases—results from Confirmatory Factor Analysis (CFA).

Factorization According to the ‘Pragmatic Approach’ (See Section 2.4 and/or Table 4)							
Factor 1: q_ht + q_af + q_hf + q_stroke + q_ckd + q_stomcanc + q_athero + q_death							
Factor 2: q_obes + q_t2dm + q_lipid							
Factor 3: q_osteo + q_hemat + q_rheum + q_brcan							
Factor 4: q_depr + q_cogn							
Diagnostics							
CFI	TLI	RMSEA	SRMR	χ^2^	df	*p*	
0.990	0.988	0.076	0.060	545.53	113	<0.001	
Standardized loadings (λ)							
Factor	Item	λ	λ SE	z	*p*	λ −95% CI	λ 95% CI
1	q_ht	0.720	0.032	22.50	<0.001	0.658	0.783
1	q_af	0.751	0.021	35.20	<0.001	0.709	0.793
1	q_hf	0.772	0.020	37.80	<0.001	0.732	0.812
1	q_stroke	0.783	0.019	42.00	<0.001	0.747	0.820
1	q_ckd	0.747	0.023	32.20	<0.001	0.702	0.793
1	q_stomcanc	0.718	0.020	36.60	<0.001	0.679	0.756
1	q_athero	0.792	0.020	39.20	<0.001	0.753	0.832
1	q_death	0.695	0.027	25.80	<0.001	0.642	0.748
2	q_obes	0.798	0.019	41.70	<0.001	0.760	0.835
2	q_t2dm	0.877	0.013	66.10	<0.001	0.851	0.903
2	q_lipid	0.830	0.017	48.90	<0.001	0.797	0.863
3	q_osteo	0.776	0.020	38.80	<0.001	0.737	0.815
3	q_hemat	0.742	0.020	36.80	<0.001	0.703	0.782
3	q_rheum	0.798	0.015	53.80	<0.001	0.769	0.827
3	q_brcan	0.824	0.014	57.70	<0.001	0.796	0.851
4	q_depr	0.849	0.015	56.10	<0.001	0.820	0.879
4	q_cogn	0.849	0.015	56.10	<0.001	0.819	0.878
Ten largest modification indices (Mis)							
lhs	op	rhs	MI	SEPC			
F3	=~	q_stomcanc	90.70	−0.498			
F1	=~	q_osteo	85.17	−0.507			
F4	=~	q_stomcanc	63.36	−0.505			
q_af	~~	q_osteo	48.44	−0.469			
F4	=~	q_osteo	47.73	−0.667			
F3	=~	q_ht	45.65	0.492			
q_hemat	~~	q_rheum	44.28	−0.395			
F4	=~	q_ht	39.18	0.558			
q_hf	~~	q_stroke	38.72	−0.418			
q_stomcanc	~~	q_brcan	36.75	−0.419			

CFI = Comparative Fit Index; TLI = Tucker–Lewis Index; RMSEA = Root Mean Square Error of Approximation; SRMR = Standardized Root Mean Square Residual; χ^2^ = chi-square test of model fit; df = degrees of freedom; *p* = *p*-value. λ = standardized factor loading (strength of association between an item and its factor); SE = standard error of λ; z = z-test statistic; CI = confidence interval for λ. Higher λ indicates stronger alignment of an item with its latent factor. Modification indices (MI) quantify the expected improvement in model fit (drop in χ^2^) if a fixed parameter were freed; SEPC = standardized expected parameter change, indicating the magnitude and direction of the potential effect. Modification indices are reported for transparency but were not applied in the final model, to avoid overfitting. The lhs (left-hand side) and rhs (right-hand side) columns denote the two model elements involved in a potential additional parameter. The column op specifies the operator: =~ indicates a factor loading (latent factor “measured by” an observed item), while ~~ indicates a covariance or residual correlation between two items or factors (i.e., the two variables [items] share unexplained variance not captured by the factors).

## Data Availability

The survey data supporting the findings of this study are not publicly available due to the fully anonymous and voluntary nature of the data collection, which prevents linking responses to individual participants. Data can be made available from the corresponding author upon reasonable request, respecting participant confidentiality.

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
