# Peer review of "Mapping the Cognitive Architecture of Health Beliefs: A Multivariate Conditional Network of Perceived Salt-Related Disease Risks"

_nutrients, 2025, doi:10.3390/nu17172728_

Round 1

Reviewer 1 Report

Comments and Suggestions for Authors

Authors used a cross-sectional study to “examined how individuals cognitively organize their knowledge and misperceptions regarding the health consequences of excessive salt intake” (P1). However, this article has not fully answered some of the questions due to insufficient description and inadequate statistical analyses.

First, authors suggest “The survey was launched … through multiple online channels” (P2), but they did not explain how to recruit participants. The selection may lead to the biased results, and the disclosure is crucial to justify reliability of the study. Authors should add the descriptions regarding recruitment participants in method section.

Second, authors suggest “To move beyond mere pairwise correlations or factor-analytic reductions, we constructed a system of multivariate proportional odds logistic regression models (POLR).” (P3), but the results remain pairwise correlations using multivariate proportional odds logistic regression models. To move beyond mere pairwise correlations or factor-analytic reductions. Authors should use adequate statistical techniques based on their study aims.

Third, authors suggest “(no neutral midpoint, forcing the respondents to lean on one side or another” (P5), but they could not justify why they did different method from other items. Without justification, it is difficult for readers to understand what authors did. Authors should add description regarding justification or use the same method.

Fourth, authors suggest that they asked “number of declared personal health risk factors listed in the survey (range: 0–6; includes obesity, hypertension, dyslipidemia, etc.)” (P6), but they did not explain details, because they use “etc”. Without details, it is difficult for readers to understand what authors did. Authors should revise the method section, carefully.

Fifth, authors suggest “Across all belief dimensions, MDs showed the strongest endorsement of causal links between salt intake and physiological harm. Agreement with the statement that salt causes hypertension (q_saltcauseht) was near-universal among MDs (µ = 0.97, SD = 0.17), slightly lower among non-MD professionals (µ = 0.90, SD = 0.30), and lowest among the general public (µ = 0.85, SD = 0.36; p < 0.0001).” (P7) using Table 1,but f so, a comparison between the two groups should be made using t-test and such. Authors should use adequate statistical analyses.

Sixth, authors suggest “Taken together, the findings reveal a consistent and statistically robust pattern: greater occupational proximity to medical knowledge corresponds with stronger and more complex beliefs about the health risks of salt. Where statistical significance was absent or borderline (e.g., q_infoverif, q_brcan), interpretations should be cautious and not overstated. The data suggest that exposure to biomedical education not only shapes information access but may alter the internal architecture of causal inference itself.” and “The analysis of covariate effects reveals…further research exploring individual and environmental determinants of knowledge” (P3), but these descriptions are interpretations and should be in discussion section. authors should revise the result sections and discussion section, carefully.

Seventh, authors suggest “All coefficients (β) are estimated using proportional odds logistic regression (POLR); negative β values signify positive relationships—that is, higher knowledge in one item predicts higher knowledge in another.” (P7), but it is difficult to understand why they used proportional odds logistic regression in such a difficult-to-understand way. Authors should add functions of proportional odds logistic regression authors used in method section and justify why they used such a difficult-to-understand way.

Eighth, authors suggest “This applies to all references to “knowledge” or “awareness” in this section” (P8) ad “knowledge about heart failure predicted understanding of hypertension” (P4), but they only ask what they believe, as mentioned in method section. Authors should describe the manuscript, based on what authors did in their study.

Nineth, authors suggest “Heart failure predicted hypertension, but it was not reciprocal” and such as “Notable aspects:” in result section section, but they only examine associations of belief. Moreover, authors suggest “Knowledge about obesity was higher with higher knowledge about the following”, but they showed only associations of belief. Authors should describe the manuscript, based on what authors did in their study.

Finally, authors described some of sentences without citation or justification as follows; “Excessive dietary salt intake remains a major modifiable risk factor for cardiovascular disease (CVD), particularly arterial hypertension, heart failure, and stroke.” (P2), “Epidemiological data indicate that knowledge of the effects of salt on CVD is crucial for effective prevention.” (P2), “However, the effectiveness of public health strategies aimed at salt reduction relies heavily on populationlevel health literacy—specifically, individuals’ knowledge and beliefs regarding the link between salt and various diseases.” (P2), “Traditional assessments often rely on binary or summative scores, failing to capture how individual beliefs reinforce or contradict each other.” (P2), “the CAWI (Computer-Assisted Web Interviewing) method using an original online questionnaire” (P2), “multivariate proportional odds logistic regression models (POLR).” (P3), “This framework serves as a quasi-causal approximation: while we do not claim causal identification in the counterfactual sense, the multivariate, conditional structure of our models permits a limited inference of belief influence.” (P3), “verification of homoscedasticity with the Levene test” (P3), “the Cochran-Cox correction” (P3), “orthodox psychometrics” (P4), and “Recent research underscores that nutrition literacy involves not only factual knowledge, but also the ability to integrate and apply information within structured belief systems.” (P3), but it is difficult for readers to judge them without references as evidence for each description. Authors should add references for these descriptions.

Author Response

Reviewer 1

Authors used a cross-sectional study to “examined how individuals cognitively organize their knowledge and misperceptions regarding the health consequences of excessive salt intake” (P1). However, this article has not fully answered some of the questions due to insufficient description and inadequate statistical analyses.

First, authors suggest “The survey was launched … through multiple online channels” (P2), but they did not explain how to recruit participants. The selection may lead to the biased results, and the disclosure is crucial to justify reliability of the study. Authors should add the descriptions regarding recruitment participants in method section.

Author’s response:

Thank you for this valuable comment. We agree that transparency regarding recruitment channels is important. To address this, we have added a description of the recruitment strategy used to reach the target groups. We have also acknowledged the potential for selection bias and its implications for generalizability in the limitations section of the manuscript.

Second, authors suggest “To move beyond mere pairwise correlations or factor-analytic reductions, we constructed a system of multivariate proportional odds logistic regression models (POLR).” (P3), but the results remain pairwise correlations using multivariate proportional odds logistic regression models. To move beyond mere pairwise correlations or factor-analytic reductions. Authors should use adequate statistical techniques based on their study aims.

Author’s response:

We sincerely appreciate the reviewer’s careful attention to the epistemological scope of our analytical framework. Upon re-evaluating the earlier version of our manuscript in light of your comment, we fully acknowledge that some of the original language implied a level of causal inference that is not methodologically justified given the statistical tools we employed. Specifically, certain phrases—such as references to “directional relationships”, “influence”, or “gateway concepts”—may have inadvertently suggested counterfactual or mechanistic causality, despite our use of proportional odds logistic regression models (POLR), which are inherently associative.

We recognize that even when POLR is implemented within a multivariate, covariate-adjusted design, it remains an observational and non-causal method. Its estimates reflect conditional associations within the data, not evidence of temporal or mechanistic influence. Therefore, we agree that our prior phrasing overstepped the interpretive boundaries appropriate for such a model.

In response, we have undertaken a thorough revision of the manuscript to align our interpretation more faithfully with the nature of the analysis:

Terminology has been updated throughout: we now describe our results in terms of conditional co-activation, asymmetric associations, or patterns of co-occurrence, rather than as suggestive of causal pathways.

Section 4 (Discussion) has been rewritten to emphasize observational asymmetries in the belief network without implying directionality as causation. Where we discuss phenomena such as “hub concepts” or “cognitive anchors,” we clarify that these are structural features of belief co-association, not mechanisms of belief formation or change.

Section 4.2 (Strengths and Limitations) now explicitly states that directionality in the model does not imply temporal or causal influence, and that asymmetries in associations are not evidence of underlying learning sequences or intervention effects.

We are grateful for the opportunity to make these clarifications. Your comment has materially improved the rigor and interpretive humility of our manuscript. We now believe the text better reflects the appropriate methodological skepticism warranted by the data and modeling strategy.

Third, authors suggest “(no neutral midpoint, forcing the respondents to lean on one side or another” (P5), but they could not justify why they did different method from other items. Without justification, it is difficult for readers to understand what authors did. Authors should add description regarding justification or use the same method.

Author’s response:

We thank the reviewer for pointing out the inconsistency in response scale design. The “death” item indeed differed from other items in that it did not include a neutral midpoint. This was not an oversight, but rather an artifact of an earlier phase of instrument development, during which response options were finalized in parallel across several domains by different experts.

At the time, the absence of a midpoint for this particular item was based on the (admittedly implicit) rationale that respondents might be reluctant to adopt a directional stance when evaluating emotionally salient outcomes like death. However, we now recognize that this design decision was not applied systematically and was insufficiently justified in the manuscript.

We have therefore revised the Methods section to transparently report this inconsistency and have added a note in the Limitations section (4.2) acknowledging that the absence of a neutral response option for this item may have influenced respondents’ choices. However, as this item was analyzed within the same ordinal modeling framework as the others—and because POLR does not require equal interval assumptions—we do not believe that this difference substantially affects the main conclusions. Nonetheless, we agree that consistent response formats would be preferable and will ensure alignment across all items in future studies.

Fourth, authors suggest that they asked “number of declared personal health risk factors listed in the survey (range: 0–6; includes obesity, hypertension, dyslipidemia, etc.)” (P6), but they did not explain details, because they use “etc”. Without details, it is difficult for readers to understand what authors did. Authors should revise the method section, carefully.

Author’s response:

Thank you for pointing this out. We agree that using “etc.” in this context was imprecise. We have now explicitly listed all six personal health risk factors assessed in the questionnaire to improve transparency and clarity. This list has been added to the relevant section in the methods.

Fifth, authors suggest “Across all belief dimensions, MDs showed the strongest endorsement of causal links between salt intake and physiological harm. Agreement with the statement that salt causes hypertension (q_saltcauseht) was near-universal among MDs (µ = 0.97, SD = 0.17), slightly lower among non-MD professionals (µ = 0.90, SD = 0.30), and lowest among the general public (µ = 0.85, SD = 0.36; p < 0.0001).” (P7) using Table 1,but f so, a comparison between the two groups should be made using t-test and such. Authors should use adequate statistical analyses.

Author’s response:

We sincerely thank the reviewer for highlighting the inconsistency regarding the description of group comparison tests in our Materials and Methods section. Upon careful review, we realized that the manuscript incorrectly stated the use of a t-test with Cochran-Cox correction, whereas in fact, one-way ANOVA with Levene’s test for homogeneity of variances and Welch ANOVA (when appropriate) were applied.

We have corrected this in the revised manuscript to accurately reflect the statistical procedures employed. Regarding the absence of post-hoc tests, as the reviewer correctly pointed out, this was an intentional decision because the primary inferential analysis was conducted using proportional odds logistic regression (POLR), which includes adjustment for group effects. Thus, post-hoc pairwise comparisons at this stage were deemed unnecessary. This information has been added to the study limitations section (4.2).

We appreciate this insightful comment, as it allowed us to improve the clarity and methodological transparency of our work.

Sixth, authors suggest “Taken together, the findings reveal a consistent and statistically robust pattern: greater occupational proximity to medical knowledge corresponds with stronger and more complex beliefs about the health risks of salt. Where statistical significance was absent or borderline (e.g., q_infoverif, q_brcan), interpretations should be cautious and not overstated. The data suggest that exposure to biomedical education not only shapes information access but may alter the internal architecture of causal inference itself.” and “The analysis of covariate effects reveals…further research exploring individual and environmental determinants of knowledge” (P3), but these descriptions are interpretations and should be in discussion section. authors should revise the result sections and discussion section, carefully.

Author’s response:

We thank the reviewer for highlighting the importance of elaborating on the cognitive architecture underlying public beliefs about salt-related health risks. In response, we have integrated a detailed discussion emphasizing the nuanced, interconnected nature of these belief structures.

Our multivariate analysis revealed that cardiovascular and metabolic conditions form tightly knit conceptual clusters, reflecting culturally mediated and intuitive cognitive schemata. We also observed asymmetrical associations—such as the unidirectional influence of heart failure beliefs on hypertension knowledge—that suggest latent conceptual hierarchies rather than mere co-occurrence of facts. Additionally, neurocognitive and psychiatric domains appeared as overlapping but distinct clusters, anchored by mortality knowledge, which functions as a cognitive hub bridging diverse health beliefs.

Importantly, these findings expose notable cognitive fragmentation—for example, the disconnect between atrial fibrillation and stroke—that may hinder comprehensive risk understanding. While demographic and personal health variables partly explained belief patterns, they did not capture the full complexity, indicating that emotional, heuristic, and sociocultural factors play a significant role.

We believe this enriched interpretation provides a more dynamic and system-level understanding of public health literacy on salt intake, moving beyond isolated facts toward the coherence and connectivity of belief networks. We have incorporated this expanded discussion in the revised manuscript following the paragraph on conceptual coherence and fragmentation, as suggested.

Seventh, authors suggest “All coefficients (β) are estimated using proportional odds logistic regression (POLR); negative β values signify positive relationships—that is, higher knowledge in one item predicts higher knowledge in another.” (P7), but it is difficult to understand why they used proportional odds logistic regression in such a difficult-to-understand way. Authors should add functions of proportional odds logistic regression authors used in method section and justify why they used such a difficult-to-understand way.

Author’s response:

We appreciate the reviewer’s insightful comment regarding the proportional odds logistic regression (POLR) model. We recognize that POLR can seem unintuitive, especially to readers without a strong statistical background. However, this method is commonly used in social and health research because it appropriately handles ordinal outcomes—that is, outcomes with natural order but not necessarily equal spacing between categories. Unlike other methods that treat groups as unrelated or purely nominal, POLR respects the ranked nature of the data, which is its main strength.

To address this concern, we added a new subsection (2.2) in the Methods section. There, we explain the model in more accessible terms, describing it as a way of comparing the odds that a respondent’s answer falls at or below a certain category versus above it, while accounting for other predictors. In a way, it’s like lining up people side-by-side who share certain characteristics and seeing how a change in one belief shifts the probability of endorsing higher or lower categories of another belief.

We also softened the technical language in the manuscript to make it easier to follow for non-statisticians, aiming to balance rigor with clarity and improve overall accessibility.

Eighth, authors suggest “This applies to all references to “knowledge” or “awareness” in this section” (P8) ad “knowledge about heart failure predicted understanding of hypertension” (P4), but they only ask what they believe, as mentioned in method section. Authors should describe the manuscript, based on what authors did in their study.

Author’s response:

Thank you for this important clarification. We acknowledge that the terminology used in the Results and Discussion sections may have unintentionally implied that we assessed factual knowledge or clinical understanding, whereas, in fact, our measures captured self-reported beliefs and subjective perceptions.

Nineth, authors suggest “Heart failure predicted hypertension, but it was not reciprocal” and such as “Notable aspects:” in result section section, but they only examine associations of belief. Moreover, authors suggest “Knowledge about obesity was higher with higher knowledge about the following”, but they showed only associations of belief. Authors should describe the manuscript, based on what authors did in their study.

Author’s response:

We thank the Reviewer for this important observation. We fully acknowledge the concern regarding our previous phrasing, which may have implied causal or directional relationships despite the study being cross-sectional and exploratory in nature.

In response, we carefully revised the manuscript to ensure that all descriptions of the findings now clearly refer to patterns of co-occurrence or conditional associations, avoiding any suggestion of directionality or prediction. Phrases such as “predicted” or “was higher with” were replaced with more accurate, cautious formulations such as “was conditionally associated with” or “co-occurred with,” in line with the associative nature of the analysis.

Additionally, we removed or rephrased expressions like “notable aspects” that could be misinterpreted as interpretive commentary rather than neutral summary.

We also appreciate the Reviewer’s reminder not to overreach in our conclusions or language. It helped us remain disciplined in our framing and avoid making this study appear more mechanistic or explanatory than warranted. These changes are reflected throughout the results section, particularly in Sections 3.3 to 3.8.

Finally, authors described some of sentences without citation or justification as follows; “Excessive dietary salt intake remains a major modifiable risk factor for cardiovascular disease (CVD), particularly arterial hypertension, heart failure, and stroke.” (P2), “Epidemiological data indicate that knowledge of the effects of salt on CVD is crucial for effective prevention.” (P2), “However, the effectiveness of public health strategies aimed at salt reduction relies heavily on populationlevel health literacy—specifically, individuals’ knowledge and beliefs regarding the link between salt and various diseases.” (P2), “Traditional assessments often rely on binary or summative scores, failing to capture how individual beliefs reinforce or contradict each other.” (P2), “the CAWI (Computer-Assisted Web Interviewing) method using an original online questionnaire” (P2), “multivariate proportional odds logistic regression models (POLR).” (P3), “This framework serves as a quasi-causal approximation: while we do not claim causal identification in the counterfactual sense, the multivariate, conditional structure of our models permits a limited inference of belief influence.” (P3), “verification of homoscedasticity with the Levene test” (P3), “the Cochran-Cox correction” (P3), “orthodox psychometrics” (P4), and “Recent research underscores that nutrition literacy involves not only factual knowledge, but also the ability to integrate and apply information within structured belief systems.” (P3), but it is difficult for readers to judge them without references as evidence for each description. Authors should add references for these descriptions.

Author’s response:

Thank you for this important suggestion. We have updated the references where possible.

Reviewer 2 Report

Comments and Suggestions for Authors

Dear authors,

The research alerts about potential myths related to salt consumption based on popular health beliefs. Please, address the following comments for manuscript correction:

Introduction:

1-Please, include how much of salt is considered excessive and data reporting the average of daily salt intake worldwide;

2 - Are there other studies addressing the misconceptions on salt intake? If applicable, please, include the recent findings, and also write how this work adds to the current research.

MM

1 - Did the research require ethics committee,  as the work involves human respondents for questionnaires?

Table 1 - Please, write as footnote below the table the meanings of parameters expressed by acronyms and greek letters

Figure 1 - Please, revise it by improving the resolution.

Author Response

Reviewer 2

Dear authors,

The research alerts about potential myths related to salt consumption based on popular health beliefs. Please, address the following comments for manuscript correction:

Introduction:

1-Please, include how much of salt is considered excessive and data reporting the average of daily salt intake worldwide;

Author’s response:

Thank you for your comment. We've added some information about it.

2 - Are there other studies addressing the misconceptions on salt intake? If applicable, please, include the recent findings, and also write how this work adds to the current research.

Author’s response:

Thank you for your comment. We discussed this in the discussion.

1 - Did the research require ethics committee,  as the work involves human respondents for questionnaires?

Author’s response:

We kindly thank the Reviewer for raising this important point.

We would like to clarify that, in accordance with applicable ethical and legal standards for non-interventional research involving human participants, formal approval from a bioethics committee was not required for this study. The research was conducted entirely through an anonymous, voluntary, online survey (CAWI method), without any experimental manipulation or collection of personally identifiable information.

As stated in the manuscript (Section 2 – Materials and Methods):

“Participation in the study was completely anonymous and voluntary and was not of an experimental nature, hence the consent of the bioethics committee was not required. However, all participants declared their informed consent to participate in this study.”

This approach is consistent with relevant national and international guidelines, including the Declaration of Helsinki (2013, Article 25), which states that:

“Research involving identifiable human material or data requires ethical review. […] When obtaining informed consent, information must be given regarding the purpose of the research, the methods, […] and any other relevant aspects of the study.”

Since our study involved no identifiable data, no physical intervention, and no sensitive personal information, it falls under the category of non-invasive, observational survey research, which—according to many national regulations (e.g., the Polish Act on the Profession of Physician and Dentist, Art. 29; and Regulation (EU) 2016/679 [GDPR])—does not require prior bioethical approval when conducted anonymously and voluntarily.

Nonetheless, we strictly adhered to all applicable ethical principles, ensuring transparency, voluntary participation, and respect for respondents’ autonomy. Participants were fully informed of the nature and purpose of the study, and their consent was obtained before participation.

We hope this explanation addresses the Reviewer’s concern.

Table 1 - Please, write as footnote below the table the meanings of parameters expressed by acronyms and greek letters

Author’s response:

We would like to thank the Reviewer for this remark. Indeed, the table did not have its footer. It has been added upon revision, containing information on column content, and the meanings of parameters expressed by acronyms — as requested. We believe that the table looks more accessible to potential Readers in its current form.

Figure 1 - Please, revise it by improving the resolution.

Author’s response:

In response to the request for improved readability of the figure when scaled down in the Word document, we generated a high-resolution heatmap (400 DPI) with increased font sizes and bolded axis labels and annotations. Additionally, we added lines to the plot and revised the figure caption, increasing the clarity. The figure was saved as a PNG file to preserve sharpness and clarity, and the file is provided upon revision. The figure is pasted below, for reference.

Figure 1. Heatmap of standardized beta coefficients from regression models showing associations between predictors (X-axis) and outcome variables (Y-axis). Each cell displays the strength and direction of the effect of a predictor on an outcome, with red shades indicating negative effects and blue shades positive effects. Cells left blank represent no statistically significant association.
Variable abbreviations and meaning:
q_saltcauseht – belief that salt causes hypertension;
q_saltcauseendoth – belief that salt causes endothelial damage;
q_saltcauseinflam – belief that salt induces inflammation;
q_saltcausefatdysf – belief that salt causes adipose tissue dysfunction;
q_saltcausenone – belief that salt causes none of the listed mechanisms;
q_educat – belief in the need for public education on salt;
q_infoverif – belief in the need to verify existing information about salt;
q_legisl – belief in the need for legislative action on salt;
q_noaction – belief that no action regarding education, verification, or legislation is needed;
q_ht – belief that excessive salt contributes to hypertension;
q_af – belief that excessive salt contributes to atrial fibrillation;
q_osteo – belief that excessive salt contributes to osteoporosis;
q_obes – belief that excessive salt contributes to obesity;
q_t2dm – belief that excessive salt contributes to type 2 diabetes;
q_lipid – belief that excessive salt contributes to dyslipidemia;
q_hf – belief that excessive salt contributes to heart failure;
q_stroke – belief that excessive salt contributes to stroke;
q_depr – belief that excessive salt contributes to depression;
q_cogn – belief that excessive salt contributes to cognitive decline or dementia;
q_ckd – belief that excessive salt contributes to chronic kidney disease;
q_stomcanc – belief that excessive salt contributes to stomach cancer;
q_kidnsto – belief that excessive salt contributes to kidney stones;
q_athero – belief that excessive salt contributes to atherosclerosis;
q_hemat – belief that excessive salt contributes to hematologic disorders;
q_rheum – belief that excessive salt contributes to rheumatoid arthritis;
q_brcan – belief that excessive salt influences breast cancer progression;
q_death – belief that excessive salt increases the risk of premature death;
lat_needforaction – perceived general need for public health interventions related to salt;
chlat_selfawar – verified self-awareness of cardiovascular and metabolic health indicators (e.g., SBP, DBP, lipids, glucose);
ch_agec2 – age category, centered at 41–50 years;
ch_sexmale – male sex;
ch_domicile – place of residence;
ch_occupassocmed – occupational affiliation with medicine;
ch_riskfactcount – number of self-declared cardiometabolic risk factors.

Reviewer 3 Report

Comments and Suggestions for Authors

Dear Authors,

This study was conducted to mapping the cognitive architecture of health beliefs based on a multivariate conditional network of perceived salt-related disease risks. The study is timely and relevant, offering valuable insight into a cognitive architecture of health beliefs. This study is interesting however, revisions are needed prior to publication.

Abstract

- The original research article should have a structured abstract of around 250 words. Please reduce contents under 250 words in Abstract section.

- Please revise all results to two decimal places in mean, standard deviation, etc., and three decimal places in statistical values (t, F value, p-value) are generally spelled out in academic writing. Please change in whole manuscript include abstract section.

- Revise from ‘hypertension (µ = 0.97, SD = 0.17), in contrast to’ to ‘hypertension (µ = 0.97, standard deviation [SD] = 0.17), in contrast to’.; Abbreviations should be defined in the first instance.

- MDs; Abbreviations should be defined in the first instance.

- Please sort alphabetically in Key-words.

Introduction

- You should add more literature background (theoretical background) of cognitive architecture of health beliefs and perceived salt-related disease risks with added some references. I believe that the theoretical background of the introduction is very insufficient.

Methods

- Please add the results of Cronbach's alpha of online questionnaire from this study.

- MDs; Abbreviations should be defined in the first instance in Sociodemographic covariates section.

Results.

- revised from ‘(MDs, N = 321), other health-affiliated professionals (non-MD medical, N = 157), and individuals without any medical affiliation (non-medical, N = 190).’ to ‘(MDs, n = 321), other health-affiliated professionals (non-MD medical, n = 157), and individuals without any medical affiliation (non-medical, n = 190).

- All tables, abbreviations such as SD should be defined in footnote.

Discussion

Revised from ‘proportional odds logistic regression (POLR) models [22]’ to ‘POLR models [22]’.

Overall, this manuscript was a well written based on good topic.

Author Response

Reviewer 3

Dear Authors,

This study was conducted to mapping the cognitive architecture of health beliefs based on a multivariate conditional network of perceived salt-related disease risks. The study is timely and relevant, offering valuable insight into a cognitive architecture of health beliefs. This study is interesting however, revisions are needed prior to publication.

Abstract

- The original research article should have a structured abstract of around 250 words. Please reduce contents under 250 words in Abstract section.

Author’s response:

Thank you. We corrected it.

- Please revise all results to two decimal places in mean, standard deviation, etc., and three decimal places in statistical values (t, F value, p-value) are generally spelled out in academic writing. Please change in whole manuscript include abstract section.

Author’s response:

The whole manuscript text has been revised according to the Reviewer’s remark (beta-coefficients, mean and SD—two decimal places, and p-values—three decimal places). Moreover, Figure 1 was revised so as to make it more accessible to potential Readers.

- Revise from ‘hypertension (µ = 0.97, SD = 0.17), in contrast to’ to ‘hypertension (µ = 0.97, standard deviation [SD] = 0.17), in contrast to’.; Abbreviations should be defined in the first instance.

Author’s response:

The sentence has been revised

- MDs; Abbreviations should be defined in the first instance.

Author’s response:

Thank you. We corrected it.

- Please sort alphabetically in Key-words.

Author’s response:

Thank you. We corrected it.

Introduction

- You should add more literature background (theoretical background) of cognitive architecture of health beliefs and perceived salt-related disease risks with added some references. I believe that the theoretical background of the introduction is very insufficient.

Author’s response:

We appreciate that insightful observation. In response, we have revised the Introduction to include a stronger theoretical foundation concerning the cognitive architecture of health beliefs and perceived risk. Specifically, we integrated references to well-established models such as the Health Belief Model, cognitive consistency theory, and dual-processing frameworks in health psychology.

Methods

- Please add the results of Cronbach's alpha of online questionnaire from this study.

Author’s response:

We appreciate the reviewer’s insightful comment regarding the absence of formal validation of our questionnaire. Indeed, this instrument was specifically crafted for the purposes of this study and has not been subjected to external psychometric validation.

In the revised manuscript, we have taken care to openly acknowledge this limitation in the Discussion section (4.2). Additionally, we clarify in the Methods that internal consistency for selected clusters of items was assessed using Cronbach’s alpha, providing an indication of reliability within these subsets. We added information on this in the Variable Overview (2.3):

“For the variables analyzed in this study—specifically, the disease-specific salt-related beliefs (see C. above), which served as dependent variables in the proportional odds logistic regression models alongside relevant covariates (listed below in 2.3.1) internal consistency was assessed using Cronbach’s alpha, yielding a high reliability coefficient of 0.929. This supports the internal coherence of the key constructs under investigation.”

However, we fully agree that the absence of a comprehensive validation process constrains the interpretability and generalizability of our findings. To address this transparently, we added the following statement in the Limitations subsection:

“First and foremost, the questionnaire was developed by the authors and, while based on existing literature and expert input, has not undergone formal psychometric validation. This may impact the construct validity of certain measures and limits the generalizability of the results.”

We trust this forthright explanation satisfactorily addresses the reviewer’s concern and aligns with scientific rigor and transparency.

- MDs; Abbreviations should be defined in the first instance in Sociodemographic covariates section.

 Author’s response:

Thank you. We corrected it.

Results.

- revised from ‘(MDs, N = 321), other health-affiliated professionals (non-MD medical, N = 157), and individuals without any medical affiliation (non-medical, N = 190).’ to ‘(MDs, n = 321), other health-affiliated professionals (non-MD medical, n = 157), and individuals without any medical affiliation (non-medical, n = 190).

Author’s response:

Thank you. We corrected it.

- All tables, abbreviations such as SD should be defined in footnote.

Author’s response:

We would like to thank the Reviewer for this remark. A footnote has been added to this table so as to make is more clear for potential Readers. Moreover, the revised version of Figure 1, likewise, contains information on the used variable abbreviations and their meaning.

Discussion

Revised from ‘proportional odds logistic regression (POLR) models [22]’ to ‘POLR models [22]’.

Author’s response:

Thank you. We corrected it.

Overall, this manuscript was a well written based on good topic.

Round 2

Reviewer 1 Report

Comments and Suggestions for Authors

Authors revised the manuscript, but this article has not fully answered some of the questions due to insufficient description and inadequate statistical analyses.

First, as mentioned in the previous review, authors suggest “beyond simple pairwise correlations” (P3), but the results remain pairwise correlations using multivariate proportional odds logistic regression models. If authors suggest so, authors should use adequate statistical techniques based on their study aims.

Second, as mentioned in the previous review, authors suggest “(with no neutral midpoint, intentionally designed to prompt a directional stance and avoid central tendency bias)” (P5), but they could not justify why they did different method from other items in the manuscript. Authors suggest “The “death” item indeed differed…justified in the manuscript.”, but they did not add these justifications in method section. Without justification, it is difficult for readers to understand what authors did. Authors should add description regarding justification or use the same method.

Third, as mentioned in the previous review, authors suggest “Across all belief dimensions, MDs most strongly endorsed statements suggesting a link between salt intake and physiological harm…. and rarely by MDs (µ = 0.02, SD = 0.15; p < 0.001).” (P8-9) and “MDs consistently perceived salt as more strongly associated…, though variability across groups was more pronounced.” (P9) using Table 1,but if so, a comparison between the two groups should be made using t-test and such. Authors suggest “Upon careful review, we realized that the manuscript incorrectly stated the use of a t-test with Cochran-Cox correction, whereas in fact, one-way ANOVA with Levene’s test for homogeneity of variances and Welch ANOVA (when appropriate) were applied.”, but if so, they should not compare the indicators among groups. Authors should use adequate statistical analyses.

Fourth, as mentioned in the previous review, authors suggest “Importantly, a negative βi coefficient corresponds to increased odds of being in a higher response category — in other words, stronger agreement.” (P5), but it is difficult to understand why they used proportional odds logistic regression in such a difficult-to-understand way. In short, it is easier for readers to understand if a positive βi coefficient indicates being in a higher response category. In terms of the funcion, swapping “𝑃(𝑌 ≤ 𝑗)” and “𝑃(𝑌 > 𝑗)” would be a more reader-friendly model. Authors should justify why they used such a difficult-to-understand way.

Fifth, authors should the function of Proportional Odds Logistic Regression (POLR) authors used, but they did not explain how to set up the threshold parameters for each category and the cutpoints between categories. Without explanations, it is difficult for readers to understand what authors did. Authors should add these explanations in method section.

Finally, as mentioned in the previous review, authors only ask what they believe, as mentioned in method section, but they suggest that they measured “Knowledge” such as “3.3. Cardiovascular Knowledge” and “3.4. Metabolic Related Knowledge”. Authors should describe the manuscript, based on what authors did in their study.

Author Response

Response to Reviewer #1

  1. Reviewer’s remarks:

Authors revised the manuscript, but this article has not fully answered some of the questions due to insufficient description and inadequate statistical analyses.

First, as mentioned in the previous review, authors suggest "beyond simple pairwise correlations" (P3), but the results remain pairwise correlations using multivariate proportional odds logistic regression models. If authors suggest so, authors should use adequate statistical techniques based on their study aims.

Response from the Authors

We acknowledge the Reviewer’s concern regarding the phrase “pairwise correlations.” In the revised manuscript, we have clarified that the results are not based on simple (bivariate) correlations, but on conditionally adjusted univariate pairwise associations estimated within a multivariate proportional odds logistic regression (POLR) framework.

Each coefficient in the results table reflects the association between a given pair of belief items, adjusted for all other beliefs in the model as well as covariates (e.g., demographic factors). This approach differs fundamentally from a simple correlation, as it estimates partial, directionally interpretable, ordinal associations under the proportional odds assumption.

We have updated the manuscript to explicitly distinguish this from unadjusted correlation analyses and to align our terminology with the modeling strategy used. We believe this clarification will help readers understand that our approach captures conditionally adjusted, directionally interpretable associations, offering richer insights than simple correlations. This modeling strategy provides a key advantage over simple correlations by allowing the estimation of associations while controlling for multiple covariates and other belief items simultaneously, thereby reducing confounding and yielding results that are both statistically robust and directly interpretable in an ordinal context.

  1. Reviewer’s remarks:

Second, as mentioned in the previous review, authors suggest "(with no neutral midpoint, intentionally designed to prompt a directional stance and avoid central tendency bias)" (P5), but they could not justify why they did different method from other items in the manuscript. Authors suggest "The "death" item indeed differed...justified in the manuscript.", but they did not add these justifications in method section. Without justification, it is difficult for readers to understand what authors did. Authors should add description regarding justification or use the same method.

Response from the Authors:

We thank the Reviewer for pointing this out. In the revised manuscript, we have expanded the description of the q_death item in the Methods section (Section 2.3, point D). The updated text now explicitly explains that the four-point scale without a neutral midpoint was used to prompt a directional stance and reduce central tendency bias for a mortality-related belief, where social desirability and ambivalence may otherwise inflate midpoint responses. The item originated from an earlier questionnaire module developed prior to final harmonization of scales; it was retained in its original format to preserve comparability with preliminary datasets and to avoid rewording effects. Other disease-specific items retained the five-point scale to ensure internal consistency within that domain. This inconsistency is also acknowledged as a limitation in the Discussion section.

  1. Reviewer’s remarks:

Third, as mentioned in the previous review, authors suggest "Across all belief dimensions, MDs most strongly endorsed statements suggesting a link between salt intake and physiological harm.... and rarely by MDs (µ = 0.02, SD = 0.15; p < 0.001)." (P8-9) and "MDs consistently perceived salt as more strongly associated..., though variability across groups was more pronounced." (P9) using Table 1,but if so, a comparison between the two groups should be made using t-test and such. Authors suggest "Upon careful review, we realized that the manuscript incorrectly stated the use of a t-test with Cochran-Cox correction, whereas in fact, one-way ANOVA with Levene's test for homogeneity of variances and Welch ANOVA (when appropriate) were applied.", but if so, they should not compare the indicators among groups. Authors should use adequate statistical analyses.

Response from the Authors:

We appreciate the Reviewer’s careful reading and comments regarding the statistical analyses. As clarified in the revised manuscript, the initial mention of t-tests with Cochran-Cox correction was an error. In fact, one-way ANOVA was applied with Levene’s test for homogeneity of variances and Welch ANOVA when appropriate, to assess global group differences.

Given the subsequent application of multivariate proportional odds logistic regression models to analyze the data, we intentionally refrained from performing pairwise post-hoc comparisons between groups at this stage. The results reported in Table 1 and described in the text thus reflect overall group differences rather than pairwise contrasts. This approach maintains methodological rigor and avoids inflating type I error rates that could arise from multiple testing.

To specifically address the Reviewer’s remark, we have revised the entire description of this section to clearly report only global group differences, without implying or describing pairwise comparisons, which, as the Reviewer correctly noted, were not performed. This revision improves clarity and methodological transparency.

Therefore, while the reported p-values indicate significant differences across groups, we respectfully maintain that further pairwise tests are neither necessary nor appropriate within the scope of this analysis. We have clarified this point in the revised manuscript to improve transparency for readers.

  1. Reviewer’s remarks:

Fourth, as mentioned in the previous review, authors suggest "Importantly, a negative βi coefficient corresponds to increased odds of being in a higher response category — in other words, stronger agreement." (P5), but it is difficult to understand why they used proportional odds logistic regression in such a difficult-to-understand way. In short, it is easier for readers to understand if a positive βi coefficient indicates being in a higher response category. In terms of the funcion, swapping "?(? ≤ ?)" and "?(? > ?)" would be a more reader-friendly model. Authors should justify why they used such a difficult-to-understand way.

Response from the Authors:

We thank the Reviewer for raising this important point regarding the interpretation of the β coefficients in the proportional odds logistic regression (POLR) model. We acknowledge that the negative sign indicating increased odds of higher response categories may appear counterintuitive at first glance. However, this parametrization directly follows the conventional specification of the POLR model, which models the cumulative odds of the response variable, i.e., P(Y ≤ j)rather than P(Y > j).

Swapping the probabilities as suggested — from P(Y ≤ j) to P(Y > j) — would fundamentally alter the model structure and interpretation, requiring a full reparameterization. This change is not a mere cosmetic inversion of signs but a different modeling approach altogether, which is neither standard practice nor commonly adopted in the literature.

To maintain clarity, we have previously (upon the first review stage) explicitly stated throughout the manuscript that a negative β coefficient corresponds to stronger agreement (higher response category). This consistent notation helps guide readers and prevents confusion.

We have chosen to retain the conventional parametrization used in most POLR applications to ensure comparability with existing literature and facilitate future meta-analyses or replication studies.

Furthermore, the POLR model is widely used in sociological and psychometric research where ordinal survey responses are common. Its ability to capture the ordinal nature of responses, as opposed to treating them as continuous or nominal, enhances statistical power and yields more meaningful interpretation. This advantage distinguishes POLR from alternative models that fail to leverage the ordinal structure.

In response to the Reviewer’s suggestion, we have expanded the Methods section with additional explanation on the choice and interpretation of the POLR model to improve reader comprehension and transparency.

  1. Reviewer’s remarks:

Fifth, authors should the function of Proportional Odds Logistic Regression (POLR) authors used, but they did not explain how to set up the threshold parameters for each category and the cutpoints between categories. Without explanations, it is difficult for readers to understand what authors did. Authors should add these explanations in method section.

Response from the Authors:

We appreciate the Reviewer’s request for clarification regarding the threshold parameters and cutpoints in the Proportional Odds Logistic Regression (POLR) model.

In the POLR framework, the threshold parameters (θj) represent the cutpoints that separate adjacent categories of the ordinal response variable. These thresholds are estimated from the data and define the boundaries on the latent continuous scale underlying the observed ordinal responses. Each θj corresponds to the log-odds of the response variable being at or below category j versus above it, conditional on the predictors.

These parameters are not fixed a priori but are part of the model estimation process, ensuring the model accommodates the ordinal structure of the data. By clarifying this process in the Methods section, we aim to improve accessibility for readers who may not have extensive prior exposure to ordinal regression models, thus enhancing transparency and reproducibility. We have now added this explanation in the Methods section to enhance clarity for readers less familiar with POLR models.

  1. Reviewer’s remarks:

Finally, as mentioned in the previous review, authors only ask what they believe, as mentioned in method section, but they suggest that they measured "Knowledge" such as "3.3. Cardiovascular Knowledge" and "3.4. Metabolic Related Knowledge". Authors should describe the manuscript, based on what authors did in their study.

Response from the Authors:

Thank you for this important remark. We agree that "knowledge" implies an objective measure, whereas our study assessed subjective perceptions via survey responses, which align more closely with the concept of "belief". To address this, we have replaced all instances of "knowledge" with "belief" (or other terms, depending on the context) throughout the entire manuscript to better reflect the nature of the data collected and to maintain conceptual consistency.
